# LILO: Bayesian Optimization with Interactive Natural Language Feedback

## Abstract

For many real-world applications, feedback is essential in translating complex, nuanced, or subjective goals into quantifiable optimization objectives. We propose a language-in-the-loop framework that uses a large language model (LLM) to convert unstructured feedback in the form of natural language into scalar utilities to conduct BO over a numeric search space. Unlike preferential BO, which only accepts restricted feedback formats and requires customized models for each domain-specific problem, our approach leverages LLMs to turn varied types of textual feedback into consistent utility signals and to easily include flexible user priors without manual kernel design. At the same time, our method maintains the sample efficiency and principled uncertainty quantification of BO. We show that this hybrid method not only provides a more natural interface to the decision maker but also outperforms conventional BO baselines and LLM-only optimizers, particularly in feedback-limited regimes.

## 1 Introduction

Bayesian optimization (BO) is a powerful strategy for optimizing expensive-to-evaluate black-box objectives (Brochu et al., 2010; Shahriari et al., 2015; Frazier, 2018). Real BO deployments, however, frequently optimize for complex tradeoffs over many outcomes, where the objective is not clearly specified, and optimization must be guided by feedback from a decision maker (DM) rather than by direct access to ground-truth utility values. Existing preference BO methods address this by learning from comparisons or ratings (González et al., 2017; Chu & Ghahramani, 2005; Lin et al., 2022; Feng et al., 2024), yet they typically require narrow feedback formats, custom likelihoods models, and careful kernel or hyper-prior engineering for the surrogate model. In contrast, black box optimization approaches driven by large language models (LLMs) end-to-end offer a more flexible interface: they can translate rich natural language input and bring in domain knowledge (Liu et al., 2024; Yang et al., 2023; Brown et al., 2020). But unlike BO, such approaches usually lack calibrated uncertainty estimates, principled exploration–exploitation tradeoffs, and reliable convergence guarantees.

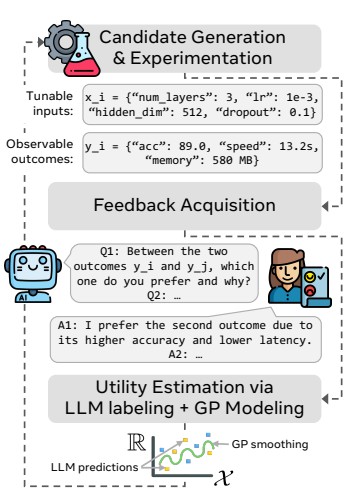

Figure 1: Core steps of LILO.

In this paper, we introduce *Language-in-the-loop Optimization* (LILO), a framework designed to combine the complementary strengths of BO and LLMs while avoiding their respective weaknesses. Rather than letting the LLM directly control the optimization, we place it in a supporting role: the LLM interprets free-form feedback and prior knowledge expressed in natural language, and translates them into structured signals for Gaussian process (GP) utility surrogates. This design leverages LLMs' strengths at understanding language and contextual reasoning, while addressing the instability, poor sample efficiency, and weak generalization that arise when relying on LLMs' raw in-context learning (ICL) abilities (Brown et al., 2020; Zhao et al., 2024; Yin et al., 2024). By mapping natural language feedback into quantitative utilities that GPs can model, LILO preserves BO's calibrated uncertainty and acquisition mechanisms. Beyond showing that such a hybrid is feasible, we provide a systematic study of the design choices that make it effective and practical.

The result is a principled, empirically validated framework that combines the expressivity of LLMs with the reliability of BO. In summary, our main contributions are as follows:

1. We propose `LILO`, a novel BO approach for settings that require eliciting utility information from a DM. `LILO` leverages natural language feedback, which can be more information-rich than pairwise comparisons or scalar ratings commonly used in other methods.

2. We show how to translate such natural language feedback into quantitative latent utilities that can be used effectively by a surrogate and acquisition function, systematically exploring the design choices required to render this approach both effective and practical.

3. We demonstrate the improved performance and sample efficiency of `LILO` compared to existing baselines on a variety of synthetic and real-world environments, outperforming both classic GP-based and LLM-based Bayesian optimization methods.

4. We show that the LLM is able to not merely serve as a "utility translator", but also successfully leverage its domain knowledge acquired during pre-training, enhancing the optimization.

## 1.1 PROBLEM DEFINITION

We consider the problem of optimizing the outputs of a black-box system with respect to preferences of a DM . Let $\mathcal{X} \subset \mathbb{R}^d$ denote the search space of controllable parameters, and let each $y \in \mathcal{Y} \subset \mathbb{R}^k$ define the outcomes of an experiment obtained through an expensive-to-evaluate function $y = f(x)$, where $f : \mathcal{X} \to \mathcal{Y}$ maps parameters to outcomes in the outcome space $\mathcal{Y}$.

A DM associates with each observed outcome $y$ a latent utility value $u = g(y) = (g \circ f)(x) \in \mathbb{R}$, where $g : \mathcal{Y} \to \mathbb{R}$ is an unknown utility function that reflects the DM's preferences. In practice, the DM is often unable to directly specify the closed form of the utility function, so the exact scalar utility value is not directly observable. Instead, the DM can provide feedback, e.g., based on pairwise comparisons (Lin et al., 2022) or, in this work, natural language, to inform about the shape of the utility function $g$. Our goal is to identify parameters $x \in \mathcal{X}$ that maximize satisfaction of the DM, i.e., to solve the composite optimization problem defined as:

$$x^* := \arg\max_{x \in \mathcal{X}} \; g(f(x)).$$

For example, as illustrated in Figure 1, in hyperparameter tuning for machine learning models, $x$ would be the collection of hyperparameters such as batch size or the learning rate and $y$ would be the model's evaluation metrics such as accuracy, training time, and memory usage. Given a specific set of outcomes $y$, the utility $u$ reflects the DM's overall satisfaction about the observed outcomes $y$.

Note that this is a highly flexible setup that can accommodate a wide variety of problems. When there is no intermediate outcome that can be observed and the preference is directly defined over the parameter space $\mathcal{X}$, it is equivalent to letting $y = f(x) = x$, reducing this problem to the classic *preferential BO* (PBO) problem (González et al., 2017; Astudillo et al., 2023). When the $y$ is high-dimensional (e.g., image or text) or involves hard-to-quantify outcomes (e.g., the decision maker's subjective feelings), the proposed method is still applicable by modeling the composite structure $(g \circ f)(x)$ directly instead of $f$ and $g$ separately and skipping explicit modeling of $y$ completely.

## 2 RELATED WORKS

**Preferential Bayesian Optimization.** Classic BO methods (Shahriari et al., 2015; Frazier, 2018) combine a probabilistic surrogate model – typically a GP – with an acquisition function that balances exploration and exploitation. *Preference learning* extends this idea to settings where explicit numerical objectives are unavailable and instead only preferential feedback (e.g. in form of pairwise or comparative judgments) can be obtained from a decision maker. The seminal work of Chu & Ghahramani (2005) introduced GP preference learning with a probit likelihood to model noisy pairwise data. González et al. (2017) formalized preferential BO, providing algorithms and guarantees for optimization from pairwise comparisons. Lin et al. (2022) extended this idea with *Bayesian optimization with preference exploration*, where DMs provide feedback on hypothetical outcomes of experiments in order to best learn the DM's utility function. This decision-theoretic policy, Expected Utility of the Best (EUBO), had been extended to the best-out-of-k selection in Astudillo et al. (2023). Connections to dueling and multi-dueling bandits, made for instance by Sui et al. (2017), highlight links between kernelized BO and bandit feedback. Together, these methods demonstrate principled approaches to preference-guided optimization, but rely on rigidly structured feedback

(pairwise or best-of-$k$ comparisons), limiting their ability to exploit natural language feedback that is both more expressive and natural for human decision makers.

**Optimization with LLMs.** Modern LLMs have been shown to be effective optimizers when used with in-context demonstrations (Yang et al., 2023). Agarwal et al. (2025) explore dynamically updating LLM prompts with past solutions including uncertainty estimates. Other works (Ramos et al., 2023; Cai et al., 2025) explore the synergy of LLMs and BO based on GP surrogates. Liu et al. (2024) investigate using LLMs for warm starting BO, surrogate hints, and proposal generation. Kristiadi et al. (2024) treat LLMs as a fixed feature extractors and train standard BO surrogates on the embeddings. Austin et al. (2024) and Handa et al. (2024) use an LLM to conduct preference elicitation to perform BO over a discrete set of candidates based on binary feedback maintaining an explicit parametric posterior of each individual candidate point. While these studies show promising results, some of their designs may present challenges for scalability, data requirements, or coherent integration between LLM and BO components, as the roles of the LLM and surrogates are often used in parallel rather than in an inter-dependent manner within a unified framework.

In black-box optimization beyond BO, Alamdari et al. (2024) use LLMs to generate synthetic prior feedback to cold-start bandits. Ye et al. (2025) propose LOLA, integrating LLM predictions with UCB bandits for online content experiments. Kobalczyk et al. (2025) improve the LLM's abilities to ask good clarifying questions by leveraging concepts of Bayesian Experimental Design (BED) – shifting the conversation to maximize information gain in interaction with the decision maker.

**LLM as Few-Shot Learners and Bayesian Predictors.** The foundation of much of this work's natural language aspect lies in the ICL abilities of LLMs. The seminal GPT-3 paper (Brown et al., 2020) showed that LLMs can act as effective few-shot learners. Multiple subsequent works take a Bayesian view on ICL. Panwar et al. (2023) investigate conditions where transformers approximate Bayes-optimal predictors, while Falck et al. (2024) test the martingale property and find systematic deviations suggesting ICL is not strictly Bayesian. Müller et al. (2023) argue that ICL reflects approximate Bayesian posteriors learned during transformer pretraining. Requeima et al. (2024) propose extracting predictive distributions directly from LLMs, making the Bayesian link explicit through natural language conditioning. Zhu & Griffiths (2024) propose an iterated ICL mechanism to approximate the implicit Bayesian priors encoded in LLMs, offering a general-purpose interpretability tool that can inform how these models' beliefs might be leveraged or mitigated in the downstream optimization tasks such as BO or preference elicitation.

## 3 BACKGROUND

In this section, we familiarize the reader with Pairwise GP models and the EUBO acquisition function which play a key role in preferential BO as well as our framework.

**Pairwise GP.** A standard model for pairwise preference data follows Chu & Ghahramani (2005), where a latent utility function $g : \mathcal{Y} \to \mathbb{R}$ is endowed with a GP prior. Given comparison pairs $\{(y_k, y_k')\}$ and binary labels $p_k \in \{0, 1\}$ indicating whether $y_k$ is preferred over $y_k'$, preferences are assumed to arise from utility differences modelled with the probit likelihood:

$$\Pr(p_k = 1 \mid g) = \Phi\left(g(y_k) - g(y_k')\right), \tag{1}$$

where $\Phi$ is the standard normal CDF, and the likelihood factorizes over comparisons. Because the probit likelihood is non-Gaussian, the posterior over latent utilities is intractable. The Laplace approximation is therefore used to obtain a Gaussian approximation. This enables tractable prediction and acquisition optimization. Further details can be found in Chu & Ghahramani (2005).

**EUBO.** In the setup of Lin et al. (2022), the goal is to learn the latent utility function $g$ via obtaining pairwise preference feedback. The Expected Utility of Best Option (EUBO) (Astudillo et al., 2023) acquisition function guides preference-exploration by selecting comparisons expected to yield the greatest increase in the utility of the best option. Concretely, consider two candidate outcome vectors $(y_1, y_2)$. The EUBO score for the pair $(y_1, y_2)$ is defined as:

$$\text{EUBO}(y_1, y_2) = \mathbb{E}\left[\max\{g(y_1), g(y_2)\}\right]. \tag{2}$$

Under a Gaussian posterior, the above expression can be computed analytically by standard results for a truncated normal differences. By selecting the pair with highest EUBO, one actively queries in a way that, in expectation, improves knowledge about top-utility options, accelerating convergence towards high-utility outcomes while minimizing preference queries.

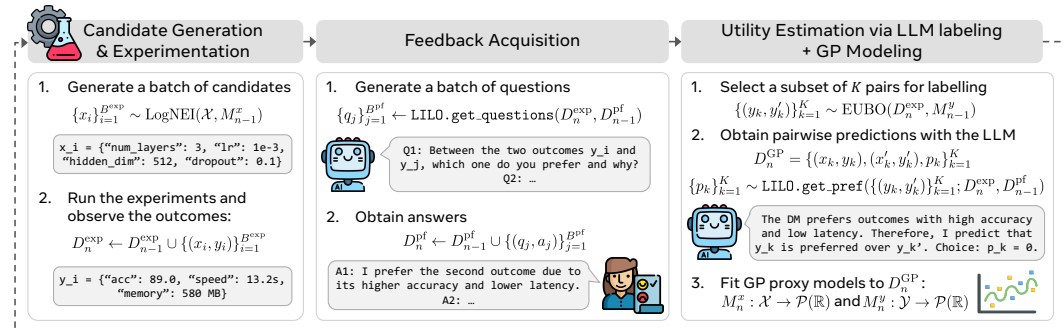

Figure 2: *Overview of* LILO: Configurations to be evaluated on the black box function are generated with a BO acquisition function leveraging the GP utility model. After observing associated outcomes, the LLM generates a set of questions for the decision maker based on all previously observed outcomes and user feedback. Responses are added to the LLM's conversation history, based on which the LLM is asked to provide preferential feedback on a set of pairwise comparisons. This feedback is then used to update the GP utility model, after which the iteration continues.

# 4 METHOD

## 4.1 LILO: LANGUAGE-IN-THE-LOOP OPTIMIZATION

The key idea behind LILO is to use an iterative preferential BO algorithm with an explicit GP-based utility model to steer the optimization in a principled fashion while harnessing the power of LLMs to extract preferences from the decision maker in the form of natural language feedback and agentically devise questions during such feedback elicitation. Figure 2 provides an overview of the optimization loop after initialization. We describe the core steps of LILO below. Algorithm 1 presents pseudo-code, with subroutines described in Appendix A.1. The exact formats of the prompts used can be found in Appendix A.2.

Our BO procedure consists of $T$ sequential trials indexed by $n \in \{1, 2, \ldots, T\}$. We define by $D_n^{\exp} = \{(x_i, y_i)\}$ the accumulated *experimental dataset* consisting of input–output pairs observed up to trial $n$. Additionally, $D_n^{\mathrm{pf}} = \{(q_j, a_j)\}$ denotes the *preference feedback dataset* that contains a history of the DM's natural language feedback in the form of answers $a_j$ to LLM-generated queries $q_j$. From these datasets, in each trial, we fit two proxy GP models (for details see below):

- $M_{n-1}^x : \mathcal{X} \to \mathcal{P}(\mathbb{R})$, approximating the composite function $g \circ f : \mathcal{X} \to \mathbb{R}$,
- $M_{n-1}^y : \mathcal{Y} \to \mathcal{P}(\mathbb{R})$, approximating the DM's utility function $g : \mathcal{Y} \to \mathbb{R}$.

**The entry point.** Before entering the main optimization loop, the algorithm begins by eliciting the DM's high-level optimization goals. At this stage, no experimental outcomes exist ($D_0^{\exp} = \varnothing$), so the LILO agent generates $B^{\mathrm{pf}}$ general questions for the DM. After obtaining the DM's answers, the preference feedback dataset is instantiated as $D_0^{\mathrm{pf}} = \{(q_j, a_j)\}_{j=1}^{B^{\mathrm{pf}}}$.

Next, LILO proceeds to the main optimization loop, in which each trial consists of three stages: 1) Candidate Generation & Experimentation, 2) Feedback Acquisition, and 3) Utility Estimation via LLM Labeling + GP Modeling.

**Candidate Generation & Experimentation.** At $n = 1$, when no proxy model has been fit yet, we generate the first batch of candidates uniformly at random. For $n > 1$, using the current input-space model $M_{n-1}^x$, we optimize an acquisition function over the search space $\mathcal{X}$ to select a batch of $B^{\exp}$ candidate inputs $\{x_i\}_{i=1}^{B^{\exp}}$. Our acquisition function of choice (although alternative options are also viable, see Appendix C.8) is the batch version of *(Log) Noisy Expected Improvement* (Ament et al., 2023), which is well-suited here because $M_{n-1}^x$ is trained on noisy, LLM-derived utility estimates rather than evaluations of the true utility function. Each selected input $x_i$ is then evaluated on the (black-box) function $f$, producing outcomes $y_i = f(x_i)$. With this, the experimental dataset becomes $D_n^{\exp} = D_{n-1}^{\exp} \cup \{(x_i, y_i)\}_{i=1}^{B^{\exp}}$.

**Feedback Acquisition.** After obtaining new experimental outcomes, the LILO agent generates $B^{\mathrm{pf}}$ questions for the domain expert to answer. These queries can include both: high-level questions regarding overall optimization goals, and questions about specific outcomes observed

---

**Algorithm 1:** `LILO`: Language-in-the-loop Optimization

---

**Require:** Max number of iterations $T$, experiment batch size $B^{\text{exp}}$, feedback batch size $B^{\text{pf}}$. **Optional:** Prior knowledge prompt $P_{\text{prior}}$

$D_0^{\text{exp}} \leftarrow \varnothing$;

$\{q_j\}_{j=1}^{B^{\text{pf}}} \leftarrow$ `LILO.get_init_questions()`;

$\{a_j\} \leftarrow$ `DM.get_answers`$(\{q_j\}_{j=1}^{B^{\text{pf}}})$;

$D_0^{\text{pf}} \leftarrow \{(q_i, a_i)\}_{i=1}^{B^{\text{pf}}}$;

**for** $n = 1$ **to** $T$ **do**

    **if** $n = 1$ **then**

        **if** $P_{prior} \neq \varnothing$ **then**

            $\{x_i\}_{i=1}^{B^{\text{exp}}} \sim$ `LILO.get_init_x`$(D_0^{\text{pf}}, P_{\text{prior}})$;

        **else**

            $\{x_i\}_{i=1}^{B^{\text{exp}}} \sim \text{Uniform}(\mathcal{X})$;

    **else**

        $\{x_i\}_{i=1}^{B^{\text{exp}}} \sim \text{LogNEI}(\mathcal{X}; M_{n-1}^x)$;

    $D_n^{\text{exp}} \leftarrow D_{n-1}^{\text{exp}} \cup \{(x_i, y_i) : y_i = f(x_i)\}_{i=1}^{B^{\text{exp}}}$;

    $\{q_j\}_{j=1}^{B^{\text{pf}}} \leftarrow$ `LILO.get_questions`$(D_n^{\text{exp}}, D_{n-1}^{\text{pf}})$;

    $\{a_j\}_{j=1}^{B^{\text{pf}}} \leftarrow$ `DM.get_answers`$(\{q_j\}_{j=1}^{B^{\text{pf}}})$;

    $D_n^{\text{pf}} \leftarrow D_{n-1}^{\text{pf}} \cup \{(q_j, a_j)\}_{j=1}^{B^{\text{pf}}}$;

    $M_n^x, M_n^y \leftarrow$ `fit_proxy_models`$(D_n^{\text{exp}}, D_n^{\text{pf}}, M_{n-1}^y)$     (See Algorithm 3 for details);

---

(which ones are preferred, how to improve them, etc.). The prompt for question generation contains all experimental outcomes and human feedback messages obtained. The LLM produces a set of $B^{\text{pf}}$ questions $q_j$, and the DM provides corresponding answers $a_j$. The preference feedback dataset becomes $D_n^{\text{pf}} = D_{n-1}^{\text{pf}} \cup \{(q_j, a_j)\}_{j=1}^{B^{\text{pf}}}$.

**Utility Estimation via LLM Labeling + GP Modeling.** To convert the natural language feedback of the DM into a usable optimization signal, we construct a dataset of $K$ pairwise preferences between outcomes in $D_n^{\text{exp}}$. The `LILO` agent is tasked with labeling these comparisons based on the DM's preference feedback in $D_n^{\text{pf}}$. We first select $K$ informative outcome pairs $\{(y_k, y_k')\}$ – using the EUBO acquisition function (Astudillo et al., 2023), which we approximate based on $M_{n-1}^y$. For each pair $(y_k, y_k')$, the LLM determines which outcome is more aligned with the DM's preferences, producing a label $p_k \in \{0, 1\}$ indicating whether or not $y_k$ is preferred over $y_k'$. This results in a dataset $D_n^{\text{GP}} = \{((x_k, y_k), (x_k', y_k'), p_k)\}_{k=1}^K$, which we use to re-fit the pairwise GP (Chu & Ghahramani, 2005) proxy models: the input-space utility model $M_n^x : \mathcal{X} \to \mathcal{P}(\mathbb{R})$ and the outcome-space utility model $M_n^y : \mathcal{Y} \to \mathcal{P}(\mathbb{R})$. These updated models will be used in the next trial to guide candidate generation and the LLM labeling process.

## 4.2 Incorporating Prior Knowledge with LILO

In addition to providing feedback on observed outcomes, decision makers with domain expertise often possess strong prior beliefs about the optimization problem at hand. These priors may include expectations about optimal parameters $x$, or insights into how specific parameters influence outcomes $y$—that is, information about the underlying mapping $f$. Conventional BO approaches, however, make it challenging to incorporate such priors, as this typically requires interpreting the decision maker's knowledge by a human and manually encoding it into the model (e.g., through specialized kernels, custom mean functions, or carefully designed priors over the surrogate model's parameters). This process can be both time-consuming and error-prone, particularly when the expert's knowledge is qualitative and difficult to formalize mathematically. A natural language interface offers a more intuitive way for decision makers to express their prior knowledge.

In scenarios where such prior information is available, we introduce a modification to `LILO` that enables warm-starting the optimization process. Inspired by Liu et al. (2024), who demonstrate that LLMs can serve as effective candidate samplers when contextual knowledge is present, we propose replacing the uniform sampling at $n = 1$ with LLM-based sampling. Specifically, given

---

**Example Conversation of `LILO` with the DM agent**

```
- DM: My goal is to keep all metrics within my thermal comfort
preferences.
- LILO: Can you provide more specific thresholds for each metric that
you consider to be within your thermal comfort preferences?
- DM: To be honest, I prefer it when the air doesn't feel drafty at my
 neck, and I don't feel too much warmth coming from the ceiling. I
also like it when there's not a big difference in temperature between
my head and ankles. As for the floor, I find it comfortable when it's
around 22-24 degrees Celsius. If it's too cold or too hot, my feet
feel uncomfortable, and that affects my overall comfort.
- LILO: How do you prioritize these metrics relative to one another?
For example, is minimizing PPD more important to you than minimizing
DR, or are they equally important?
- DM: For me, it's really important that I don't feel drafty or too
warm from the ceiling. If I had to choose, I'd say minimizing those
feelings is a bit more important than the other factors. But, at the
same time, I also value not having a big temperature difference
between my head and ankles. It's all about finding a balance, I
suppose. If one thing is off, it can really affect how I feel overall.
[...]
```

---

Example 1: Example fragments of feedback elicitation conversation of `LILO` with the DM in one of the test problem (Thermal Comfort Type A). `LILO` is able to elicit and incorporate such qualitative feedback about optimization effectively as shown in our experiments in Section 5.

$D_0^{\text{pf}}$ and additional prior information represented in textual form as $P_{\text{prior}}$, we generate $B^{\text{pf}}$ candidate points $x_i$ via LLM prompting (see Prompt 6, Appendix A.2). Incorporation of prior knowledge is optional in `LILO`; in our experiments in Section 5.3.3 we demonstrate how leveraging a good prior can significantly enhance optimization performance.

## 5 EXPERIMENTS

### 5.1 BENCHMARKING SETUP

**Test problems.** We evaluate `LILO` on synthetic and real-world outcome functions $f$, subject to various utility functions $g$. In all our test problems we let $\mathcal{X}$ be a hypercube inside $\mathbb{R}^d$ and $\mathcal{Y} \subseteq \mathbb{R}^k$. Our main benchmark considers four outcome functions (one fully synthetic, and three simulating real-world outcomes):

- *DTLZ2* (Deb et al. (2002), $d = 4, k = 8$) is a synthetic outcome function commonly used as a benchmark for multi-objective optimization algorithms.
- *Vehicle Safety* (Tanabe & Ishibuchi (2020), $d = 5, k = 3$), a simulator of a vehicle's mass and two safety-defining outcomes. The inputs are the thickness of the frontal structure of a car, which affect the vehicle's mass and safety.
- *Car Cab Design* (Liao et al. (2008), $d = 7, k = 9$), a simulator of a side-impact car crash test. The inputs measure the thickness of a car's structure, which influence the vehicle's mass, the physical impact on the passenger, and the physical impact on the car.
- *Thermal Comfort* (Fanger (1970); ISO7730 (2005), $d = 8, k = 5$) models perceptible thermal conditions and human dissatisfaction levels based on a set of environmental parameters that need to be optimized.

We consider several utility functions on the outcomes of these test problems:

- *piecewise linear*, modeling diminishing returns when outcomes reach their desired thresholds.
- *beta products*, describing bounded monotonic utilities that smoothly vary between increasing and decreasing marginal gains with respect to their inputs.
- *L1 distance*, measuring the L1 distance of outcomes from a pre-defined optimum point.
- For the Thermal Comfort problem, we consider two personas with different preferences: *Type A*) an office worker with light clothing and a moderate tolerance for different conditions; *Type B*) a summer athlete, wearing light sport kit, with a low tolerance for adverse conditions.

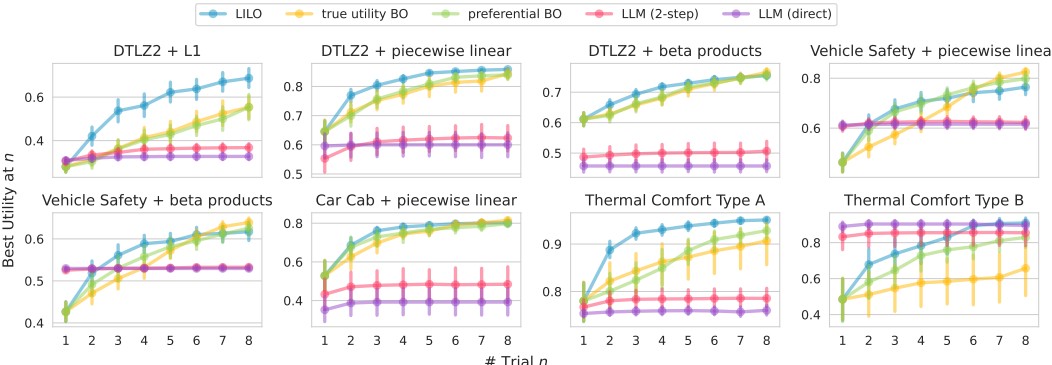

Figure 3: Maximum value of the ground-truth utility achieved after $n$ iterations. Error bars are 95% confidence intervals of the mean max value across 30 replications.

All test problems and outcome/utility combinations are described in detail in Appendix B. All utility functions are designed to take values in $[0, 1]$.

**Baselines.** We compare `LILO` against the following baselines (full details for all baselines are provided in Appendix B): ▶ **true utility BO**: a GP-based BO that directly observes ground-truth utilities $u_i = g(y_i)$ for each outcome $y_i$. Since utilities are never directly accessible in practice, this baseline serves as an oracle "gold standard". ▶ **preferential BO**: a setup with binary pairwise comparison feedback instead of natural language feedback. Pairwise comparisons between two outcomes $y, y'$ are derived from the ground-truth utility differences $g(y) - g(y')$. Similarly to true utility BO, this can be seen as a "gold standard" baseline, as in practice, pairwise comparison feedback is noisy due to inconsistencies in human decision-making. ▶ **LLM (2-step)**: an ablation of our method in which the GP model and acquisition function are replaced by LLM-based candidate generation. Specifically, in each iteration, the LLM first estimates utilities for previously observed outcomes $D_n^{\mathrm{exp}}$ (via pairwise comparisons) and then selects the next candidate batch through in-context learning. The prompt for candidate generation contains the dataset $D_n^{\mathrm{exp}}$ with the utility estimates based on the $M_n^y$ model and the history of conversation with the DM from $D_n^{\mathrm{pf}}$. This baseline resembles the LLAMBO method of Liu et al. (2024). The key difference lies in the fact that in the original setup of LLAMBO, the ground-truth utilities of individual candidates $x$ are directly observable. Here, they must be estimated based on the natural language feedback. ▶ **LLM (direct)**: a fully end-to-end LLM approach, where new candidates are proposed directly based on the observed $(D_n^{\mathrm{pf}}, D_n^{\mathrm{exp}})$ omitting the intermediate utility estimation step.

**Simulation environment.** We run the BO loop for $T = 8$ iterations, setting $B^{\mathrm{exp}} = d$ (the dimension of the search space). For the main experiments we use a feedback batch size of $B^{\mathrm{pf}} = 2$ (see Appendix C.2 for an ablation across varying values of $B^{\mathrm{pf}}$). The true utility BO baseline receives true utility observations for $B^{\mathrm{pf}}$ outcomes selected from $D_n^{\mathrm{exp}}$. The preferential BO baseline receives pairwise comparison feedback for $B^{\mathrm{pf}}$ outcome pairs selected from $D_n^{\mathrm{exp}}$. In both methods, we use the EUBO acquisition function to select the data points on which the quantitative feedback is provided (see Appendix C.5 for an ablation on the choice of the feedback acquisition function). For methods involving natural language feedback, answers to questions posed by the LLM agent are simulated with another LLM containing a textual description of the ground-truth utility function in the prompt (see Appendix B.3 for details). In the main paper, both the DM and `LILO` agents are instantiated with the Llama-3.3-70b-instruct language model. Results for other choices of LLM models are presented in Appendix C.1.

## 5.2 KEY RESULTS

We first present the results on a set of 8 environments, where no prior knowledge about the problem is provided ($P_{\mathrm{prior}} = \varnothing$), except for the variable names in the Vehicle Safety, Car Cab Design, and Thermal Comfort environments. For fairness of comparison with the quantitative baselines, we use `LILO` with random initialization, as initialization leveraging the LLM's world knowledge may give a significant performance boost as we later show in Section 5.3.3

Figure 3 shows the maximum ground-truth utility achieved after $n$ trials of experimentation and feedback collection, that is, $\max_{x \in D_n^{\mathrm{exp}}} g(f(x))$. We observe that `LILO` consistently outperforms the baselines, especially early on during the optimization. In some environments, where outcomes

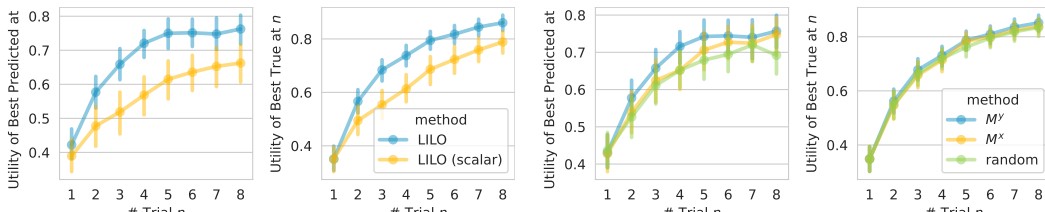

(a) Performance of `LILO` with pairwise vs. scalar utility estimation.

(b) Comparison of methods for LLM pairwise preference labeling.

Figure 4: *Comparison of `LILO` variants.* Average results over three DTLZ2 environments (30 replications per environment, values min-max standardized within an environment before aggregation).

correspond to semantically meaningful quantities, the two LLM baselines show good zero-shot performance, but in general, they fail to improve meaningfully over the course of the optimization. This observation aligns with the in-context learning literature on its diminishing return over number of examples provided (Brown et al., 2020; Zhao et al., 2024; Yin et al., 2024). Remarkably, `LILO` also substantially outperforms the true utility and preferential BO baselines – this is due to the fact that **natural language feedback can convey much richer information about the overall DM's preferences than only localized, point-specific feedback**. The DM may provide auxiliary feedback not only on the performance of specific outcomes, but also on the overall shape of their underlying utility function (e.g., the directionality of the utility function with respect to different metrics, their relative importance, etc.). See Example 1 and Appendix D for example conversations from the benchmark. With increasing number of experimental trials, the advantage of natural language feedback diminishes and scalar or pairwise utility feedback baselines catch up.

## 5.3 ADDITIONAL STUDIES

In this section, we present additional studies providing insights into the key elements of `LILO`.

### 5.3.1 PAIRWISE COMPARISONS VS. DIRECT UTILITY ESTIMATION.

The default utility estimation step in `LILO` relies on LLM-generated pairwise comparisons. As an alternative, we consider directly prompting the LLM to output scalar utility values. In this variant, instead of labeling pairwise preferences, the LLM produces scalar predictions $\hat{u}_i \in [0,1]$ for each $(x_i, y_i) \in D_n^{\text{exp}}$, resulting in a dataset $D_n^{\text{GP}} = \{(x_i, y_i, \hat{u}_i)\}$. This dataset is then used to fit proxy models $M_n^x$ and $M_n^y$ as standard GPs. All other parts of the pipeline remain unchanged. The prompt used for this method and the modified algorithm are presented in Appendix A.

**Results.** To compare the two approaches, we conducted an ablation study on the three DTLZ2 environments with varying utility functions. Figure 4a reports average results over 30 replications per environment. The right pane shows the maximum ground-truth utility achieved until the $n$-th iteration. On the left, we report the ground-truth utility of the best candidate according to the proxy model $M_n^y$, i.e. $g(\hat{x}_n^*)$, where $\hat{x}_n^* = \arg\max_{x \in D_n^{\text{exp}}} M_n^y(f(x))$. In practice, it may be preferable or even necessary to use the utility model to do this "best point selection"; especially with many observations, it will be impractical for the DM to compare a large slate of options at once – in fact, they may not be able to do this well due to mental overload. The results demonstrate that **pairwise comparisons provide more reliable utility estimates than direct scalar predictions, leading to improved optimization performance**. This observation for LLMs mirrors findings in human preference elicitation, where pairwise comparisons have been shown to yield more consistent and accurate judgments than absolute scalar ratings (Phelps et al., 2015; Hoeijmakers et al., 2024).

### 5.3.2 THE CHOICE OF PAIRS FOR LLM LABELING.

During the utility estimation step of `LILO`, we prompt the agent to label a subset of $K$ pairwise comparisons between outcomes from $D_n^{\text{exp}}$, $\{(y_k, y_k')\}_{k=1}^K$. This set of outcomes is selected by computing the value of the EUBO acquisition functions with respect to the $M_{n-1}^y$ model, and selecting the top $K$ pairs with the highest EUBO values. In our experiments, we use $K = 64$. In this section, we ablate this step by comparing it against random pair selection, and against a method that uses the $M_{n-1}^x$ instead of the $M_{n-1}^y$ model. For the latter, the EUBO is computed on pairs of inputs $(x_k, x_k')$ and the top $K$ pairs of corresponding outcomes $(y_k, y_k')$ are passed to the LLM for labeling.

**Results.** Figure 4b shows the results obtained on the three DTLZ2 environments. We observe that the use of the $M_{n-1}^y$ model for pair selection with respect to EUBO improves the predictive performance of the outcome-space proxy in identifying the best candidate (left pane). This improvement, however, does not directly translate to significant differences in terms of the optimization performance measured with respect to the ground-truth utility function (right pane). Thus, if the outcome space $\mathcal{Y}$ is high dimensional (e.g., text or image), and the mapping $\mathcal{Y} \rightarrow \mathbb{R}$ hard to model, fitting the outcome space GP models may be omitted and we may simply select points for labeling using $M_{n-1}^x$ or even randomly, simplifying the pipeline.

### 5.3.3 INCORPORATING PRIOR KNOWLEDGE.

As described in Section 4.2, LILO can also incorporate domain priors to boost optimization performance. In the following, we demonstrate this empirically, considering three types of prior messages $P_{\text{prior}}$: ▶ **point**: A message providing a sample candidate with high expected utility: *"Based on my experience, the following inputs should bring good results: $\{x\}$"*. The promising candidate is generated by sampling uniformly at random $N$ candidates, computing their ground-truth utilities, and randomly sampling a single point from the top $q\%$ of data points. ▶ **area**: A message providing expected bounds of good candidates: *"Based on my experience, inputs within these ranges should bring good results: $\{bounds\}$"*. The bounds are computed by sampling uniformly at random $N$ candidates, computing their ground-truth utilities, and taking the 0.25 and 0.75 quantiles of the top $q\%$ of data points, as the lower and upper bounds, respectively. ▶ **domain**: A message contextualizing the input parameters and the outputs. This is applicable only to semantically-meaningful environments: Vehicle Safety, Car Cab Design and Thermal Comfort. Exact forms of these messages are provided in Appendix B.4. For the DTLZ2 outcome function, we apply the point and area priors. For point knowledge we use $q = 10$ and for area knowledge we use $q = 5$ with $N = 5000$ in both cases. For Vehicle Safety, Car Cab Design and Thermal Comfort we apply the domain priors.

**Results.** Figure 5 presents a comparison of LILO 's performance with and without prior knowledge integrated into the optimization pipeline. As expected, **incorporating prior knowledge through LLM-based initialization substantially improves the starting point, resulting in better overall optimization performance**. For a more detailed breakdown of the results, see Appendix C.4. We also note that point and area knowledge types depend on the accuracy of externally provided information about $(g \circ f)$ and, in principle, could be incorporated into conventional BO pipelines using model-based approaches. However, LILO 's success with domain priors relies on the contextual understanding provided by the LLM's pre-training, which would be challenging to replicate with standard model-based methods. This ability to incorporate various types of prior knowledge in a unified fashion further underscores LILO 's flexibility.

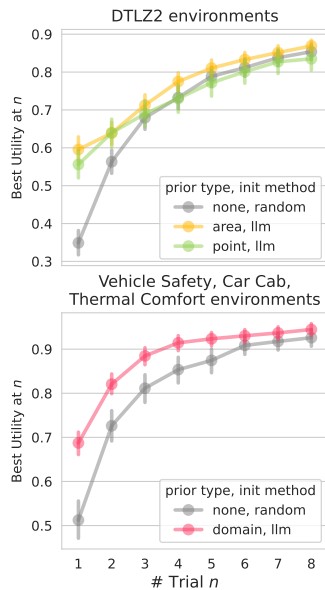

Figure 5: LILO with and without prior knowledge.

### 5.3.4 ADDITIONAL EXPERIMENTAL RESULTS

Beyond the results described above, we further conduct a series of additional experiments providing further insights. These include ablation studies on the choice of LLM models (Appendix C.1), the choice of the batch size of the feedback queries (Appendix C.2, C.3), a qualitative visualization of the fitted utility functions (Appendix C.6), and a performance analysis of LILO on longer optimization horizons (Appendix C.7).

## 6 DISCUSSION AND CONCLUSION

We presented LILO, a novel algorithm for language-in-the-loop optimization that enables decision makers to perform interactive black-box optimization by communicating their goals and preferences in natural language. This addresses a key challenge in settings where optimization objectives are not explicitly available, but where classic preference learning approaches (e.g. using pairwise feedback) are not sample-efficient enough. LILO demonstrates compelling performance in our benchmarks, and has been successfully applied in multiple real-world optimization campaigns involving complex tradeoffs between multiple metrics in online A/B testing at a large internet company.

The reason of `LILO`'s superior performance is multi-faceted. Natural language feedback is far more information-rich than other common forms of preference elicitation such as pairwise comparison as demonstrated by the experimental results where `LILO` achieves very competitive optimization results with just a few feedback queries. On the other hand, LLM-based optimization methods are fit to handle natural language feedback through ICL yet, their performance can plateau rather quickly. This phenomenon is observed throughout our experimentation where LLM-based methods' optimization traces start relatively high in the first iteration and then stop progressing meaningfully throughout the rest of the experiment. `LILO` is able to overcome such stagnation by keeping the valid probabilistic modeling and principled acquisition mechanism of BO while leveraging LLMs for its human-friendly natural language interface, agentic feedback acquisition, and domain knowledge (from both the DM as well as the LLM).

While our results are strong, several promising directions remain. The fidelity of the LLM estimator is ultimately limited by the model's ability to interpret nuanced feedback. Our benchmarks in Appendix C.1 show robustness across models, with even moderately sized ones yielding meaningful gains. Future work could explore how scaling, fine-tuning, or architectural advances further enhance optimization quality. Hybrid approaches that combine unstructured natural language with structured quantitative feedback are another possibility to mitigate the limits of ICL in the utility estimation step of `LILO`.

Moreover, LLM-based estimators may occasionally encode preferences or priors too strongly, risking subtle bias toward the model's own world knowledge rather than the intended optimization target. In our experiments, we found this effect to be manageable by ablating over a variety of utility functions and test problem combinations. Yet, future work could explore calibration strategies or adaptive weighting schemes that better balance the advantages of informative model priors with fidelity of the provided feedback.

Although we validated `LILO` extensively using both synthetic and real optimization problems, understanding its behavior across a broader range of tasks and feedback sources remains an exciting area. Evaluating performance with different natural language agents, including but not limited to human decision makers, can shed light on the robustness and generalization of the proposed method.

Looking forward, these directions open opportunities for advancing language-in-the-loop optimization toward even more adaptive, trustworthy, and broadly applicable optimization systems across diverse domains.

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

# A    DETAILS ON THE METHOD

## A.1    PSEUDO CODE

Algorithm 2 presents a pseudo code of LILO. The `fit_proxy_models` subroutine is presented in Algorithm 3.

The algorithm for LILO with scalar utility estimation is identical to LILO with pairwise preference labeling except for the `fit_proxy_models` subroutine, which is replaced with point-wise utility estimation as a scalar value in $[0, 1]$. The exact procedure is presented in Algorithms 4. Prompts used in the LILO.xxx subroutines are presented in section A.2.

---

**Algorithm 2:** LILO: Language-in-the-Loop Optimization

---

**Input:** Max number of iterations $T$, experiment batch size $B^{\text{exp}}$, feedback batch size $B^{\text{pf}}$.
      **Optional:** Prior knowledge prompt $P_{\text{prior}}$

$D_0^{\text{exp}} \leftarrow \varnothing$;
\# Ask the DM about their optimization goals
$\{q_j\}_{j=1}^{B^{\text{pf}}} \leftarrow$ LILO.get_init_questions();
$\{a_j\} \leftarrow$ DM.get_answers$(\{q_j\}_{j=1}^{B^{\text{pf}}})$;
$D_0^{\text{pf}} \leftarrow \{(q_i, a_i)\}_{i=1}^{B^{\text{pf}}}$;
**for** $n = 1$ **to** $T$ **do**
    |   \# Sample a batch of candidates;
    |   **if** $n = 1$ **then**
    |     |   **if** $P_{prior} \neq \varnothing$ **then**
    |     |       $\{x_i\}_{i=1}^{B^{\text{exp}}} \sim$ LILO.sample_init_candidates$(D_0^{\text{pf}}, P_{\text{prior}})$;
    |     |   **else**
    |     |       $\{x_i\}_{i=1}^{B^{\text{exp}}} \sim$ Uniform$(\mathcal{X})$;
    |   **else**
    |       $\{x_i\}_{i=1}^{B^{\text{exp}}} \sim$ LogNEI$(\mathcal{X}; M_{n-1}^x)$;
    |   \# Run experiments and update the experimental dataset
    |   $D_n^{\text{exp}} \leftarrow D_{n-1}^{\text{exp}} \cup \{(x_i, y_i) : y_i = f(x_i)\}_{i=1}^{B^{\text{exp}}}$;
    |   \# Generate a batch of questions
    |   $\{q_j\}_{j=1}^{B^{\text{pf}}} \leftarrow$ LILO.get_questions$(D_n^{\text{exp}}, D_{n-1}^{\text{pf}}, \{y_i\}_{i=1}^{2B^{\text{pf}}})$;
    |   \# Obtain human answers
    |   $\{a_j\}_{j=1}^{B^{\text{pf}}} \leftarrow$ DM.get_answers$(\{q_j\}_{j=1}^{B^{\text{pf}}})$;
    |   \# Update the preference feedback dataset
    |   $D_n^{\text{pf}} \leftarrow D_{n-1}^{\text{pf}} \cup \{(q_j, a_j)\}_{j=1}^{B^{\text{pf}}}$;
    |   \# Fit proxy models
    |   $M_n^x, M_n^y \leftarrow$ fit_proxy_models$(D_n^{\text{exp}}, D_n^{\text{pf}}, M_{n-1}^y)$;

---

## A.2    LILO PROMPTS

In all prompts, `experiment_data` is a markdown-formatted table of outcomes from $D_n^{\text{exp}}$. `human_feedback` is the series of questions and answer stored in $D_n^{\text{pf}}$.

In prompt 2 for question generation, `selected_outcome_indices` is the set of $2B^{\text{pf}}$ indices of selected outcomes with the feedback acquisition function (EUBO).

In prompts 3, 4, `human_feedback_summary` is a summary of `human_feedback` self-generated by LILO using the prompt presented in 5. We empirically found that including this self-summarization step brings slight improvements to the resulting estimates of the LLM.

In prompt 3, `pair_str` is a markdown-formatted table with two rows indexed by `option_0` and `option_1`.

---

**Algorithm 3:** LILO `fit_proxy_models` subroutine

---

**Input:** Experimental dataset $D^{\text{exp}}$, Feedback dataset $D^{\text{pf}}$, optional preference model $M^y$,
number of pairs for labeling $K = 64$.
```
# Generate all possible combinations of outcome pairs
```
$N \leftarrow |D^{\text{exp}}|$
$\{(y_i, y_i')\}_{i=1}^{\frac{N(N-1)}{2}} \leftarrow$ `get_outcome_pair_combinations`$(D^{\text{exp}})$
**if** $M^y = \textit{None}$ **then**
$\quad \lfloor \{(y_k, y_k')\}_{k=1}^K \leftarrow$ `random_sample`$(\{(y_i, y_i')\}_{i=1}^{\frac{N(N-1)}{2}}, \texttt{size} = K)$
**else**
$\quad$ **for** $i = 1$ **to** $\frac{N(N-1)}{2}$ **do**
$\quad\quad \lfloor v_i \leftarrow \text{EUB\^{O}}((y_i, y_i'), M^y)$
$\quad$ `indices` $\leftarrow$ `argsort`$(\{v_i\}_{i=1}^{\frac{N(N-1)}{2}})[-K:]$ // Get indices of top $K$ EUBO
$\quad\quad$ values
$\quad \lfloor \{(y_k, y_k')\}_{k=1}^K \leftarrow \{(y_i, y_i') : i \in \texttt{indices}\}$
```
# Label the selected pairs with LILO
```
**for** $k = 1$ **to** $K$ **do**
$\quad \lfloor p_k \leftarrow$ `LILO.get_pairwise_pref`$(y_k, y_k', D^{\text{exp}}, D^{\text{pf}})$
$D^{GP,y} \leftarrow \{(y_k, y_k', p_k)\}_{k=1}^K$
$D^{GP,x} \leftarrow \{(x_k, x_k', p_k)\}_{k=1}^K$ // $x_k, x_k'$ are the inputs corresponding to
$\quad y_k, y_k'$
$M^y \leftarrow$ `fit_pairwise_gp`$(D^{GP,y})$
$M^x \leftarrow$ `fit_pairwise_gp`$(D^{GP,x})$
**return** $M^x, M^y$

---

**Algorithm 4:** LILO (scalar) `fit_proxy_models` subroutine

---

**Input:** Experimental dataset $D^{\text{exp}}$, Feebdback dataset $D^{\text{pf}}$.
$N \leftarrow |D^{\text{exp}}|$
$\{\hat{u}_i\}_{i=1}^N \leftarrow$ `LILO.estimate_utilities`$(\{y_i\}_{i=1}^N; D^{\text{exp}}, D^{\text{pf}})$
$D^{GP,y} \leftarrow \{(y_i, \hat{u}_i)\}_{i=1}^N$
$D^{GP,x} \leftarrow \{(x_i, \hat{u}_i)\}_{i=1}^N$ $x_i$'s are the inputs corresponding to $y_i$'s
$M^y \leftarrow$ `fit_simple_gp`$(D^{GP,y})$
$M^x \leftarrow$ `fit_simple_gp`$(D^{GP,x})$
**return** $M^x, M^y$

---

### A.3 GP MODELING AND ACQUISITION FUNCTIONS

In our implementation of the algorithm we rely on the BoTorch Python library (Balandat et al., 2020) to implement the subroutines of GP model fitting, acquisition function evaluation and candidate generation. Specifically, proxy GP models are instances of `PairwiseGP` or `SingleTaskGP` classes, with their default settings.

**The Pairwise GP.** To model pairwise preferences, we use a GP with a probit likelihood, following Chu & Ghahramani (2005). Given a set of observed outcomes $(y_k, y_k')$ and binary preference labels $p_k \in \{0, 1\}$, the latent utility function $g(\cdot)$ is modeled as a GP. The probability that $y_k$ is preferred over $y_k'$ is given by $\Phi(g(y_k) - g(y_k'))$, where $\Phi(\cdot)$ is the cumulative distribution function of the standard normal distribution. Since the probit likelihood is non-Gaussian, we use the Laplace approximation to perform inference. This approach enables principled modeling of uncertainty and efficient learning from pairwise comparisons. For further details, we refer the readers to Chu & Ghahramani (2005).

When querying the LLM to provide pairwise labels or scalar utility estimates, we use multiple LLM samples for more reliable estimates. In our simulations, the number of LLM calls per each labeling

instance is set to 5. We use all LLM predictions (with replicates) to fit the GP proxy models enabling the proxy models to capture the predictive uncertainty of the LLM in its decision.

**(Log) Noisy Expected Improvement.** Our acquisition function of choice for candidate generation is the (Log) Noisy Expected Improvement. We the batched version of this acquisition function, which jointly evaluates the utility of an entire batch via Monte Carlo sampling—rather than selecting points greedily. Specifically, we employ the `qLogNoisyExpectedImprovement` method from BoTorch, which follows the work of Ament et al. (2023).

**EUBO.** For paired outcome selection in LLM labeling step we use the Expected Utility of Best Option acquisition function (Lin et al., 2022). In this step, we use a greedy approach for selecting a batch of $K$ pairs. For a pair of outcomes $(y_k, y'_k)$ EUBO has an explicit analytic form. We thus employ the `AnalyticExpectedUtilityOfBestOption` from BoTorch to compute its value for all possible combination of pairs and select the set of $K$ pairs with the highest utilties (see Algorithm 3).

---

**Initial question generation**

```
You are an expert in determining whether a human decision maker (DM)
is going to be satisfied with a set of experimental outcomes y = {
y_names}.

## Human feedback messages:
We have also received the following messages from the DM:

{human_feedback}

## Your task:
Given the above your task is to predict the probability of the
decision maker being satisfied with the experimental outcomes.

In order to better understand the decision maker's utility function
you want to ask them about their optimization goals.

Provide a list of questions you would ask the decision maker to better
 understand their internal utility model.

Return your final answer a a json file with the following format
containing exactly {n_questions} most important questions:
```json
{{
    "q1" : <question1>,
    ...
    "q{n_questions}" : <question{n_questions}>
}}
```
```

---

Prompt 1: The prompt used for question generation in the `LILO.get_init_questions` subroutine.

---

**Question generation**

```
You are an expert in determining w whether a human decision maker (DM)
 is going to be satisfied with a set of experimental outcomes y = {
y_names}.

## Experimental outcomes:
So far, we have obtained the following experimental outcomes:

{experiment_data}

## Human feedback messages:
We have also received the following messages from the DM:

{human_feedback}

## Your task:
Given the above your task is to predict pairwise preferences between
experimental outcomes.

In order to better understand the decision maker's utility function
you want to ask them about their optimization goals or for feedback
regarding specific experimental outcomes.

First, analyse the decision maker's goals and feedback messages to
understand their overall preferences.
Then, provide a list of questions you would ask the decision maker to
better understand their internal utility model.
Your questions can be either general or referring to specific outcomes
. For instance, you may ask the decision maker:
- questions clairfying the optimzation objective,
- to rank two (or more) outcomes,
- how to improve certain outcomes,
- for a likert-scale rating regarding a specific outcome,
- etc.
When referring to specific outcomes, always state the arm_index
involved.
Your questions should help you predict pairwise preferences between
any two experimental outcomes from the set of experimental outcomes
provided above.

Return your final answer a a json file with the following format
containing exactly {n_questions} most important questions:
```json
{{
    "q1" : <question1>,
    ...
    "q{n_questions}" : <question{n_questions}>
}}
```

Prompt 2: The prompt used for question generation in the `LILO.get_questions` subroutine.

864
865
866
867
868
869
870
871
872
873
874
875
876
877
878
879
880
881
882
883
884
885
886
887
888
889
890
891
892
893
894
895
896
897
898
899
900
901
902
903
904
905

---

**Pairwise comparisons**

```
You are an expert in determining whether a human decision maker (DM)
is going to be satisfied with a set of experimental outcomes y = {
y_names}.

## All experimental outcomes:

{experiment_data}

## Human feedback messages:
We have also received the following messages from the DM:

{human_feedback}

{human_feedback_summary}

## Your task:
Given a pair of outcomes--option_0 and option_1, your goal is to
decide which one is more preferable according to the DM's preferences.

{pair_str}

Provide your prediction as a json file with the following format:
```json
{{
    "reasoning": "Your reasoning about the DM's preferences and
option_0 vs. option_1. Do not insert new lines in your reasoning.",
    "answer" : 0 or 1
}}
```
where in "answer" you should return 0 if option_0 is preferred, or 1
if option_1 is preferred.
Return just the json file (with the header ```json), nothing else.
```

---

906
907
908
909
910
911
912
913
914
915
916
917

Prompt 3: The prompt used for pairwise comparison labeling used in the `LILO.get_pairwise_pref` subroutine.

> ### Scalar utility estimation
>
> ```
> You are an expert in determining whether a human decision maker (DM)
> is going to be satisfied with a set of experimental outcomes y = {
> y_names}.
>
> ## Experimental outcomes:
> So far, we have obtained the following experimental outcomes:
>
> {experiment_data}
>
> ## Human feedback messages:
> We have also received the following messages from the DM:
>
> {human_feedback}
>
> {human_feedback_summary}
>
> ## Your task:
> Given the above your task is to predict the probability of the
> decision maker being satisfied with the experimental outcomes.
>
> First, analyse the human feedback messages to understand the DM's
> preferences.
> Then, provide your predictions for all y's in the set of all
> experimental outcomes above.
> Return your final answer as a jsonl file with the following format:
>
> ```jsonl
> {{
>     "arm_index": "{idx0}",
>     "reasoning": <reasoning>,
>     "p_accept": <probability>
> }}
> {{
>     "arm_index": "{idx1}",
>     "reasoning": <reasoning>,
>     "p_accept": <probability>
> }}
> ...
> {{
>     "arm_index": "{idxn}",
>     "reasoning": <reasoning>,
>     "p_accept": <probability>
> }}
> ```
> Where <reasoning> should be a short reasoning for your prediction and
> <probability> should be your best estimate for the probability between
>  0 and 1 that the DM will be satisfied with the corresponding outcome.
>
> Provide your predictions for ALL y's in the set of experimental
> outcomes above. That is, for EACH outcome from {idx0}. to {idxn}.
> Do not generate any Python code. Just return your predictions as plain
>  text.
> ```

Prompt 4: The prompt used for scalar utility estimation used in the `LILO.estimate_utilities` subroutine.

**Human Feedback Summarization**

```
You are an expert in determining whether a human decision maker (DM)
is going to be satisfied with a set of experimental outcomes y = {
y_names}.

## Experimental outcomes:
So far, we have obtained the following experimental outcomes:

{experiment_data}

## Human feedback messages:
We have also received the following messages from the DM:

{human_feedback}

## Your task:
Given the above your task is to summarize the human feedback messages
into a clear description of the DM's optimization goals.
Make your summary as quantitative as possible so that it can be easily
 used for utility estimation.

After analysis the human feedback messages, return your final answer
as a json file with the following format:
```json
{{
    "summary": <summary>
}}
```
Remember about the ```json header!
```

Prompt 5: The prompt used for generating the human_feedback_summary by LILO for pairwise comparisons or scalar utility estimation.

Candidate Generation $n = 1$, LILO with prior knowledge

```
You are performing optimization of a utility function u(x) = g(y) = g(
f(x)), where x is a vector of parameters: x = {x_names} and y = f(x) =
 {y_names} is a vector of outcomes.
Each dimensions of x is in the range [0, 1].
Your goal is to find the parameters x that maximize the utility.

## Prior knowledge:
You have obtained the following prior knowledge about the experiment:
{prior_knowledge}

## Human feedback messages:
You have also received the following messages from the DM:
{human_feedback}

## Your task:
Given the above your task is the generate a set of {n_candidates}
candidate parameters x for the next round of experimentation.

First, analyse the information above, then return your final answer as
 a json file with the following format:
```json
{{
    "0": <candidate0>,
    "1": <candidate1>,
    ...
    "{n}": <candidate{n}>,
}}
```
Where each <candidatei> is a list of the candidate parameter values in
 [0, 1].
Do not write a python code for candidate generation. Just return the
required json.
Do not add any comments to your json. Remember about the ```json
header.
```

Prompt 6: The prompt used for candidate generation by LILO at $n = 1$ when prior knowledge is available (LILO.sample_init_candidates subroutine).

## B  SIMULATION SETUP

We evaluate `LILO` and the baselines on synthetic and real-world outcome functions as well as several utility functions. In all our test problems we have $\mathcal{X} = \mathbb{R}^d$ and $\mathcal{Y} = \mathbb{R}^k$. The main benchmark considers three outcome functions: *DTLZ2* ($d = 4, k = 8$, Deb et al. (2002)), *Vehicle Safety* ($d = 5, k = 3$, Tanabe & Ishibuchi (2020)), and *Car Cab Design* ($d = 7, k = 9$, Liao et al. (2008)). These test problems are matched with several utility functions: piecewise linear, beta products, and the L1 distance. All outcome and utility functions are described in detail in this section.

In our simulations, we run the BO loop for $T = 8$ iterations, setting $B^{\text{exp}} = d$ (the dimension of the search space). At $n = 1$ we hard-code the first message of the DM agent to seed the conversations. These messages are utility-dependent and presented in section B.2.

### B.1  OUTCOME FUNCTIONS

**DTLZ2**  The DTLZ2 function was introduced by Deb et al. (2002), allowing for arbitrary input dimension $d$ and output dimension $k$ subject to $d > k$. $\mathcal{X} = [0, 1]^d$. For a DLTZ2 function $f$ with a $k$-dimensional output and $d$-dimensional input, we have:

$$m := d - k + 1$$

$$h(x) := \sum_{i=m}^{d-1}(x_i - 0.5)^2$$

$$f_j(x) = -(1 + h(x))\left(\prod_{i=1}^{k-j-1}\cos\left(\frac{\pi}{2}x_i\right)\right)\mathbb{1}_{j>1}\sin\left(\frac{\pi}{2}x_{k-j-1}\right)$$

In our experiments we use $d = 8$ and $k = 4$.

**Vehicle Safety**  This a test problem for optimizing vehicle crash-worthiness with $d = 5$ and $k = 3$. $\mathcal{X} = [1, 3]^5$. We refer the readers to Tanabe & Ishibuchi (2020) for details on function definition. We normalize each component of $y = f(x)$ to lie between 0 and 1 based on empirical bound on the outcome space $\mathcal{Y}$.

**Car Cab Design**  We refer the readers to Liao et al. (2008) for details. Note that in the original problem, there are stochastic components which we replace with deterministic components fixed at their original mean values in order to obtain a deterministic ground-truth outcome function. We normalize each component of $y = f(x)$ to lie between 0 and 1 based on empirical bound on the outcome space $\mathcal{Y}$.

**Thermal Comfort**  The problem setting follows the ISO 7730 and ASHRAE 55 models, which predict human thermal sensation and dissatisfaction based on six core factors: air temperature, mean radiant temperature, humidity, air speed, clothing insulation, and metabolic rate (Fanger, 1970; ISO7730, 2005; ASHRAE55, 2020). From these, outcome measures such as Predicted Percentage Dissatisfied (PPD), draft risk (DR), vertical air temperature difference, radiant temperature asymmetry, and floor surface temperature are derived, each with threshold values associated with acceptable comfort. The goal of the optimization agent is to find environmental parameters that minimize discomfort and keep all outcomes within desirable ranges, reflecting realistic expectations of the occupant rather than arbitrary synthetic functions. This setting is widely used in thermal comfort research and can be visualized via the CBE Thermal Comfort Tool (Tartarini et al., 2020). In our implementation, the outcome function has two fixed, non-optimizable parameters: clothing insulation (clo $\in [0.3, 1.2]$) and metabolic rate (met $\in [1.0, 2.0]$) which differ for the two versions of the environments considered in this paper, as detailed in the next section.

### B.2  UTILITY FUNCTIONS

**L1 distance**  We consider a utility function which is the L1 distance from a pre-specified point $y_{\text{opt}}$. This choice of the utility function mimics the scenario where the DM wishes to keep the outcomes close to a specific desirable state.

For DTLZ2, we set $y_{\text{opt}} = [0.8, 1.0, 0.7, 1.25]$.

The message seeding the conversation takes the following form:

> **Goal message (L1 distance)**
>
> ```
> My goal is to bring all the outcome metrics as close to {opt_y} as
> possible.
> ```

**Beta products** Prior work on preference learning has utilized the Beta CDF to form utility functions. The Beta CDF provides a convenient, bounded monotonic transform that smoothly varies between increasing and decreasing marginal gains with respect to their inputs. Our utility function takes the following form:

$$g(y; \alpha, \beta) = \prod_{i=1}^{k} \text{BetaCDF}(y_i; \alpha_i, \beta_i)$$

For the DTLZ2 outcome function we set:

$$\alpha = [0.5, 2.0, 2.0, 2.0]$$
$$\beta = [0.5, 1.0, 2.0, 5.0]$$

For the Vehicle Safety outcome function we set:

$$\alpha = [0.5, 1.0, 1.5]$$
$$\beta = [1, 2, 3]$$

For this utility function, the message seeding the conversation takes the following form:

> **Goal message (beta products)**
>
> ```
> My goal is to bring all the outcome metrics as close to 1 as possible.
> Results are strongest only when every metric is high -- if any metric
> is low, it significantly reduces the overall performance.
> ```

**Piecewise linear** Analogously to Lin et al. (2022) we also consider a piecewise linear utility function. Its shape corresponds to diminishing marginal returns on outcomes and sharp declines in utility when desired thresholds are note met. For a $k$-dimensional input vector $y$, this utility function is defined as:

$$g(y; \beta_1, \beta_2, t) = \sum_{i=1}^{k} h_i(y_i),$$

where

$$h_i(y_i) = \begin{cases} \beta_{1,i} y_i + (\beta_{1,i} - \beta_{2,i}) t_i & \text{if } y_i < t_i \\ \beta_{2,i} y_i & \text{if } y_i \geq t_i \end{cases}.$$

For the DTLZ2 problem, we set

$$\beta_1 = [4.0, 3.0, 2.0, 1.0]$$
$$\beta_2 = [0.4, 0.3, 0.2, 0.1]$$
$$t = [1.0, 0.8, 0.5, 0.5]$$

For the Vehicle Safety problem, we set

$$\beta_1 = [2, 6, 8]$$
$$\beta_2 = [1, 2, 2]$$
$$t = [0.5, 0.8, 0.8]$$

For the Car Cab Design problem, we set

$$\beta_1 = [7.0, 6.75, 6.5, 6.25, 6.0, 5.75, 5.5, 5.25, 5.0, 4.75, 4.5]$$
$$\beta_2 = [0.5, 0.4, 0.375, 0.35, 0.325, 0.3, 0.275, 0.25, 0.225, 0.2, 0.175]$$
$$t = [0.64, 0.68, 0.96, 0.88, 1.06, 0.65, 0.84, 0.86, 0.58, 0.7, 0.53]$$

Here, thresholds $t_i$ correspond to the 0.75 quintiles of the outcome values $y_i$. The seeding message takes the following form:

> **Goal message (piecewise linear)**
>
> ```
> My goal is to achieve the following thresholds in each outcome {t}.
> Improvements over the thresholds are always good, but less important
> than bringing the outcomes to their threshold values. The further away
>  an outcome is from its threshold, the higher is its negative impact
> on the overall performance.
> ```

**Thermal Comfort** Our utility maps the outcome vector $Y = [\text{PPD}, \text{DR}, dT_{\text{vert}}, dT_{\text{pr}}, T_{\text{floor}}]$ to a scalar $U \in [0, 1]$ via per–outcome desirabilities that enforce being "within range", with using the Derringer-Such desirability functions. For the four "smaller is better" outcomes (PPD, DR, $dT_{\text{vert}}$, $dT_{\text{pr}}$) we use a one–sided acceptable band with a comfort threshold $L$ and an unacceptable threshold $H$ and define

$$
d_{\text{small}}(y; L, H, s) = \begin{cases} 1, & y \leq L, \\ \left( \dfrac{H - y}{H - L} \right)^s, & L < y < H, \\ 0, & y \geq H, \end{cases}
$$

so values at or below $L$ are fully desirable, values beyond $H$ are unacceptable, and values in between taper smoothly with shape $s \geq 1$. For floor temperature $T_{\text{floor}}$ we target a comfort band $[l, h]$ and tolerate a wider band $[l_{\min}, h_{\max}]$ by

$$
d_{\text{band}}(t; l, h, l_{\min}, h_{\max}, s) = \begin{cases} 1, & l \leq t \leq h, \\ \left( \dfrac{t - l_{\min}}{l - l_{\min}} \right)^s, & l_{\min} < t < l, \\ \left( \dfrac{h_{\max} - t}{h_{\max} - h} \right)^s, & h < t < h_{\max}, \\ 0, & t \leq l_{\min} \text{ or } t \geq h_{\max}. \end{cases}
$$

The overall utility is the geometric mean of the five desirabilities,

$$
U(Y) = \left( d_{\text{small}}(\text{PPD}) \cdot d_{\text{small}}(\text{DR}) \cdot d_{\text{small}}(dT_{\text{vert}}) \cdot d_{\text{small}}(dT_{\text{pr}}) \cdot d_{\text{band}}(T_{\text{floor}}) \right)^{1/5},
$$

.

We consider two versions of this utility functions with varying comfortable ranges of the outcome metrics

**Type A**. These settings are meant to simulate preferences of an office worker in light clothing and a moderate tolerance for different conditions.

$$
l_{\text{PPD}} = 0.0, \; h_{\text{PPD}} = 30.0,
$$
$$
l_{\text{DR}} = 10.0, \; h_{\text{DR}} = 35.0,
$$
$$
l_{\text{dT}_{\text{vert}}} = 3.0, \; h_{\text{dT}_{\text{vert}}} = 9.0,
$$
$$
l_{\text{dT}_{\text{pr}}} = 5.0, \; h_{\text{dT}_{\text{pr}}} = 22.0,
$$
$$
l_{\min, \text{T}_{\text{floor}}} = 16.0, \; l_{\text{T}_{\text{floor}}} = 19.0, \; h_{\text{T}_{\text{floor}}} = 26.0, \; h_{\min, \text{T}_{\text{floor}}} = 30.0.
$$

In Thermal Comfort Type A the clothing and metabolic rate parameters of the outcome function are set to clo $= 0.61$ and met $= 1.0$, respectively.

**Type B**. These settings are meant to simulate preferences of a summer athlete wearing light sport kit, with a lower tolerance for adverse conditions.

$$l_{\text{PPD}} = 0.0, \ h_{\text{PPD}} = 24.0,$$
$$l_{\text{DR}} = 30.0, \ h_{\text{DR}} = 45.0,$$
$$l_{\text{dT}_{\text{vert}}} = 2.5, \ h_{\text{dT}_{\text{vert}}} = 6.0,$$
$$l_{\text{dT}_{\text{pr}}} = 4.0, \ h_{\text{dT}_{\text{pr}}} = 12.0,$$
$$l_{\text{min},\text{T}_{\text{floor}}} = 19.0, \ l_{\text{T}_{\text{floor}}} = 20.0, \ h_{\text{T}_{\text{floor}}} = 23.0, \ h_{\text{min},\text{T}_{\text{floor}}} = 25.0.$$

In Thermal Comfort Type B the clothing and metabolic rate parameters of the outcome function are set to clo $= 0.3$ and met $= 2.0$, respectively.

The seeding message takes the following form:

---
**Goal message (Thermal Comfort)**

```
My goal is to keep all metrics within my thermal comfort preferences.
```
---

### B.3 LLM-BASED SIMULATION OF THE HUMAN PREFERENCE FEEDBACK

Evaluating black box optimization algorithm rigorously is challenging due to the replications required to discern performance in the presence of the inherent variance of the evaluation of the optimization traces. This is exacerbated in our setting where the algorithm is based on feedback from human decision makers. To be able to evaluate LILO rigorously, we therefore simulate the human preference feedback with another LLM.

For all our experiments we use Llama-3.3-70b-instruct as the language model for the human feedback simulator. A high-level representation of the prompt used to generate the answers is presented in prompt 7. `utility_func_desc` is a textual description of the specific utility function. `outcomes_markdown` is a markdown-formatted table of outcomes $y_i \in D^{\text{exp}}$ and their corresponding pre-computed ground-truth utilities $g(y_i)$. `questions_str` are the questions $\{q_j\}_{j=1}^{B^{\text{pf}}}$ generated by LILO. Finally `utility_constraints` contain additional utility-specific instructions for the LLM to generate human-like feedback and to not reveal explicitly the exact functional form of the utility, ensuring the generated answers sound natural.

### B.4 PRIOR KNOWLEDGE MESSAGES $P_{\text{PRIOR}}$

Below we present the prior messages $P_{\text{prior}}$ used in our experiments from section 5.3.3.

---
**DTLZ2 (point knowledge)**

```
- Based on my experience, the following inputs should bring good
results: {promising_point}.
```
---

---
**DTLZ2 (area knowledge)**

```
- Based on my experience, inputs within these ranges should bring good
 results {bounds}:
```
---

---
**Vehicle Safety (domain knowledge)**

```
- y_1 measures the reduction in vehicle's mass, y_2 measures the
reduction in integration of acceleration between two time points, y_3
measures the reduction in toe board intrusion in the offset-frontal
crash.
- The parameters x measure the thickness of five reinforced members
around the frontal structure of a car, which can significantly affect
the crash safety.
```
---

---

**Car Cab Design (domain knowledge)**

```
- A car is subjected to a side-impact crash test. The outcome
variables y measure the following:
    - The effect of the side-impact on a dummy is measured in terms of
 head injury, load in abdomen, pubic symphysis force, viscous
criterion (V * C), and rib deflections at the upper, middle and lower
rib locations.
    - The effect on the car are considered in terms of the vehicle's
weight, the velocity of the B-Pillar at the middle point and the
velocity of the front door at the B-Pillar.
- The parameters x describe some design aspects of the car. An
increase in dimension of the car parameters may improve safety, but
with a burden of an increased weight of the car. These parameters are
and their ranges are:
    x1: Thickness of B-Pillar inner [0.5, 1.5]
    x2: Thickness of B-Pillar reinforcement [0.45, 1.35]
    x3: Thickness of floor side inner [0.5, 1.5]
    x4: Thickness of cross members [0.5, 1.5]
    x5: Thickness of door beam [0.875, 2.625]
    x6: Thickness of door beltline reinforcement [0.4, 1.2]
    x7: Thickness of roof rail [0.4, 1.2]
- NOTE: The presented values of outcomes y represent the reduction in
mass, forces, velocities etc. So the goal is to increase y_1, ...,
y_11, corresponding to lowering the vehicle's weight and minimizing
the impact on the dummy and the car.
```

---

### B.5 BASELINES

#### B.5.1 QUANTITATIVE BASELINES

The quantitative baseline methods follow an analogous procedure to `LILO`, where the Q&A natural language feedback is replace with either scalar values of the utilities associated with a batch of outcomes (true utility BO) or pairwise comparisons between outcomes based on their ground-truth utilities (preferential BO). This feedback is obtained on $B^{\text{pf}}$ outcomes $y_i$ or paired outcomes $(y_i, y_i')$ sampled with the EUBO acquisition function. To extend predictions to the entire data set $D_n^{\text{exp}}$, we fit a simple / pairwise GP model $M_n^y : \mathcal{Y} \to \mathcal{P}(\mathbb{R})$. Subsequently a proxy model $M_n^x : \mathcal{X} \to \mathcal{P}(\mathbb{R})$ is fit to the predictions of $M_n^y$.

We present the exact implementation of these methods in algorithms 5 and 6.

#### B.5.2 LLM 2-STEP AND DIRECT

We also implement two versions of an LLM-based approach to candidate generation for optimization.

**LLM (2-step)** follows the same algorithm 2 as `LILO` with the following exceptions. In the candidate generation step, instead of using the LogNEI acquisition function, we prompt the LLM to generate a set of candidates using prompt 8.

**LLM (direct)** Omits the step of utility estimation and generates the candidates directly based on the raw human feedback and observed inputs and outcomes from $D_n^{exp}$. The prompt used is presented in prompt 9

---

---

**DM's answer generation**

```
Suppose you are a decision maker evaluating the results of a multi-
objective optimization problem.

You are given a set of multi-dimensional outcomes y = {y_names}

{utility_func_desc}

You have observed the following outcomes with their corresponding
utility values and contributions to the overall utility.

## Outcomes:

{outcomes_markdown}

The utility values are on a scale [0, 1], where (1 - very satisfied,
0.5 - neutral, 0 - very dissatisfied).

Based on the above information, provide answers to the following
questions:

## Questions:

{questions_str}

Return your final answer as a json file with the following format:
```json
{{
    "q1" : <answer to q1>,
    ...
    "q{n_questions}" : <answer to q{n_questions}>
}}
```
Before providing your final answers, analyze the shape of the utility
function in light of the questions posed.
In your final answers, you cannot reveal the explicit formula of the
utility function.
The form and the values of the utility functions is a "latent" feature
 of the human expert, thus you should not refer to it explicitly or
even mention its existence.
{utility_constraints}
State your answers in the first person (you are the decision maker).
Avoid vacuous statements.
```

Prompt 7: The prompt used for answer generation in the `DM.get_answers` subroutine.

---

**Candidate Generation, LLM (2-step)**

```
You are performing optimization of a utility function u(x) = g(y) = g(
f(x)), where x is a vector of parameters: x = {x_names} and y = f(x) =
 {y_names} is a vector of outcomes.
Each dimensions of x is in the range [0, 1].
Your goal is to find the parameters x that maximize the utility.

## Experimental Outcomes
So far, you have also observed the following inputs x and their
estimated utilities:

{experiment_data}

## Human feedback messages:
We have also received the following messages from the DM:

{human_feedback}

## Your task:
Given the above your task is the generate a set of {n_candidates}
candidate parameters x for the next round of experimentation.
Your candidates should maximize the expected improvement over the
current best candidate x^* = {x_star} with utility u(x^*) = {u_star}.

First, analyse the information above, then return your final answer as
 a json file with the following format:
```json
{{
    "0": <candidate0>,
    "1": <candidate1>,
    ...
    "{n}": <candidate{n}>,
}}
```
Where each <candidatei> is a list of the candidate parameter values in
 [0, 1].
Do not write a python code for candidate generation. Just return the
required json.
Do not add any comments to your json. Remember about the ```json
header.
```

---

Prompt 8: The prompt used for candidate generation by the LLM (2-step) baseline. In the above, `experiment_data` is a markdown formatted table of outcomes $y_i$ and their estimate utilities via the LLM-based proxy model $M^y_{n-1}(y_i)$. `x_star` and `u_star` are determined based on the estimated utilities (not ground truth $g(y_i)$'s as these are latent, non-observable quantities).

---

**Candidate Generation, LLM (direct)**

```
You are performing optimization of a utility function u(x) = g(y) = g(
f(x)), where x is a vector of parameters: x = {x_names} and y = f(x) =
 {y_names} is a vector of outcomes.
Each dimensions of x is in the range [0, 1].
Your goal is to find the parameters x that maximize the utility.

{experiment_data}

## Human feedback messages:
We have also received the following messages from the DM:

{human_feedback}

## Your task:
Given the above your task is the generate a set of {n_candidates}
candidate parameters x for the next round of experimentation.
First, analyze the human feedback messages to understand the DM's
preferences.
Then, generate a set of {n_candidates} candidate parameters x, trading
-off exploration and exploitation.
Return your final answer as a json file with the following format:
```json
{{
    "0": <candidate0>,
    "1": <candidate1>,
    ...
    "{n}": <candidate{n}>,
}}
```
Where each <candidatei> is a list of the candidate parameter values: {
x_names}, each in [0, 1].
Do not write a python code for candidate generation. Just return the
required json.
Do not add any comments to your json.
```

Prompt 9: The prompt used for candidate generation in the LLM (direct) baseline. `experiment_data` is a markdown-formatted table of inputs and outcomes. `human_feedback` is the set of questions and answers from $D_{n-1}^{pf}$.

---

**Algorithm 5:** True utility BO

---

**Input:** Max number of iterations $T$, Experiment batch size $B^{\text{exp}}$, Feedback batch size $B^{\text{pf}}$.

$D_0^{\text{exp}} \leftarrow \varnothing$

$D_0^{\text{pf}} \leftarrow \varnothing$

**for** $n = 1$ **to** $T$ **do**

    \# Sample a batch of candidates

    **if** $n = 1$ **then**

        $\{x_i\}_{i=1}^{B^{\text{exp}}} \sim \text{Uniform}(\mathcal{X})$

    **else**

        $\{x_i\}_{i=1}^{B^{\text{exp}}} \sim \text{LogEI}(\mathcal{X}; M_{n-1}^x)$

    \# Run the experiments and update the experimental dataset

    $D_n^{\text{exp}} \leftarrow D_{n-1}^{\text{exp}} \cup \{(x_i, y_i) : y_i = f(x_i)\}_{i=1}^{B^{\text{exp}}}$

    \# Sample a batch of outcomes for feedback

    **if** $n = 1$ **then**

        $\{y_i\}_{i=1}^{B^{\text{pf}}} \sim \texttt{random\_sample}(D^{exp})$

    **else**

        $\{y_i\}_{i=1}^{B^{\text{pf}}} \sim \text{EUBO}(D_n^{exp}, M_{n-1}^y)$

    \# Update the preference feedback dataset

    $D_n^{\text{pf}} \leftarrow D_{n-1}^{\text{pf}} \cup \{(y_j, u_j) : u_j = g(y_j)\}_{j=1}^{B^{\text{pf}}}$

    \# Fit a Y->U GP

    $M_n^y \leftarrow \texttt{fit\_simple\_gp}(D_n^{\text{pf}})$

    \# Label all experimental datapoints

    $D_n^{\text{GP}} \leftarrow \{(x_i, y_i, \hat{u}_i) : (x_i, y_i) \in D_n^{\text{exp}}\}$ // $\hat{u}_i$'s are the mean predictions

        with respect to $M_n^y : \mathcal{Y} \to \mathcal{P}(\mathbb{R})$

    \# Fit the proxy model

    $D_n^{\text{GP},x} \leftarrow \{(x_i, \hat{u}_i) : (x_i, y_i, \hat{u}_i) \in D_n^{\text{GP}}\}$

    $M_n^x \leftarrow \texttt{fit\_simple\_gp}(D_n^{GP,x})$

---

**Algorithm 6:** Preferential BO

**Input:** Max number of iterations $T$, Experiment batch size $B^{\text{exp}}$, Feedback batch size $B^{\text{pf}}$.

$D_0^{\text{exp}} \leftarrow \varnothing$,

$D_0^{\text{pf}} \leftarrow \varnothing$

**for** $n = 1$ **to** $T$ **do**

    # Sample a batch of candidates

    **if** $n = 1$ **then**

        $\lfloor \ \{x_i\}_{i=1}^{B^{\text{exp}}} \sim \text{Uniform}(\mathcal{X})$

    **else**

        $\lfloor \ \{x_i\}_{i=1}^{B^{\text{exp}}} \sim \text{LogNEI}(\mathcal{X}; M_{n-1}^x)$

    # Run the experiments and update the experimental dataset

    $D_n^{\text{exp}} \leftarrow D_{n-1}^{\text{exp}} \cup \{(x_i, y_i) : y_i = f(x_i)\}_{i=1}^{B^{\text{exp}}}$

    # Sample a batch of paired outcomes for feedback

    **if** $n = 1$ **then**

        $\lfloor \ \{(y_i, y_i')\}_{i=1}^{B^{\text{pf}}} \sim \texttt{random\_sample}(D^{exp})$

    **else**

        $\lfloor \ \{(y_i, y_i')\}_{i=1}^{B^{\text{pf}}} \sim \text{EUBO}(D_n^{exp}, M_{n-1}^y)$

    # Update the preference feedback dataset

    $D_n^{\text{pf}} \leftarrow D_{n-1}^{\text{pf}} \cup \{(y_j, y_j', p_j) : p_j = \mathbb{1}\{g(y_j) > g(y_j')\}\}_{j=1}^{B^{\text{pf}}}$

    # Fit a Y->U GP

    $M_n^y \leftarrow \texttt{fit\_pairwise\_gp}(D_n^{\text{pf}})$

    # Label all experimental datapoints

    $D_n^{\text{GP}} \leftarrow \{(x_i, y_i, \hat{u}_i) : (x_i, y_i) \in D_n^{\text{exp}}\}$ // $\hat{u}_i$'s are the mean predictions

        with respect to $M_n^y : \mathcal{Y} \to \mathcal{P}(\mathbb{R})$

    # Fit the proxy model

    $D_n^{\text{GP},x} \leftarrow \{(x_i, \hat{u}_i) : (x_i, y_i, \hat{u}_i) \in D_n^{\text{GP}}\}$

    $M_n^x \leftarrow \texttt{fit\_simple\_gp}(D_n^{GP,x})$

## C  ADDITIONAL EXPERIMENTAL RESULTS

### C.1  LLM ABLATION STUDY

We compare the performance of LILO depending on the choice of the LLM model. We compare the following models: Llama-3.3-70b-instruct, Llama-4-scout-17b-16e-instruct, and Qwen-3-14b. The LLM used to simulate human feedback remains set to Llama-3.3-70b-instruct across the comparisons.

We run the ablation study for the following 3 environments: the DTLZ2 outcome function, combined with the L1, beta products and piecewise linear utility functions. As in the main benchmark, we set $B^{\text{pf}} = 2$ and $B^{\text{exp}} = d = 8$.

**Results.** Table 1 presents the results. We observe that LILO performs similarly across all three LLMs, demonstrating that the success of our method is agnostic to the choice of a specific language model. We were unable to test the performance of LILO with smaller language models (e.g. LLama-3.1-8B or Qwen-3-8b) due to difficulties in ensuring that the LLM's outputs follow the required json structure, leading to parsing errors.

### C.2  PREFERENCE FEEDBACK BATCH SIZE ABLATION

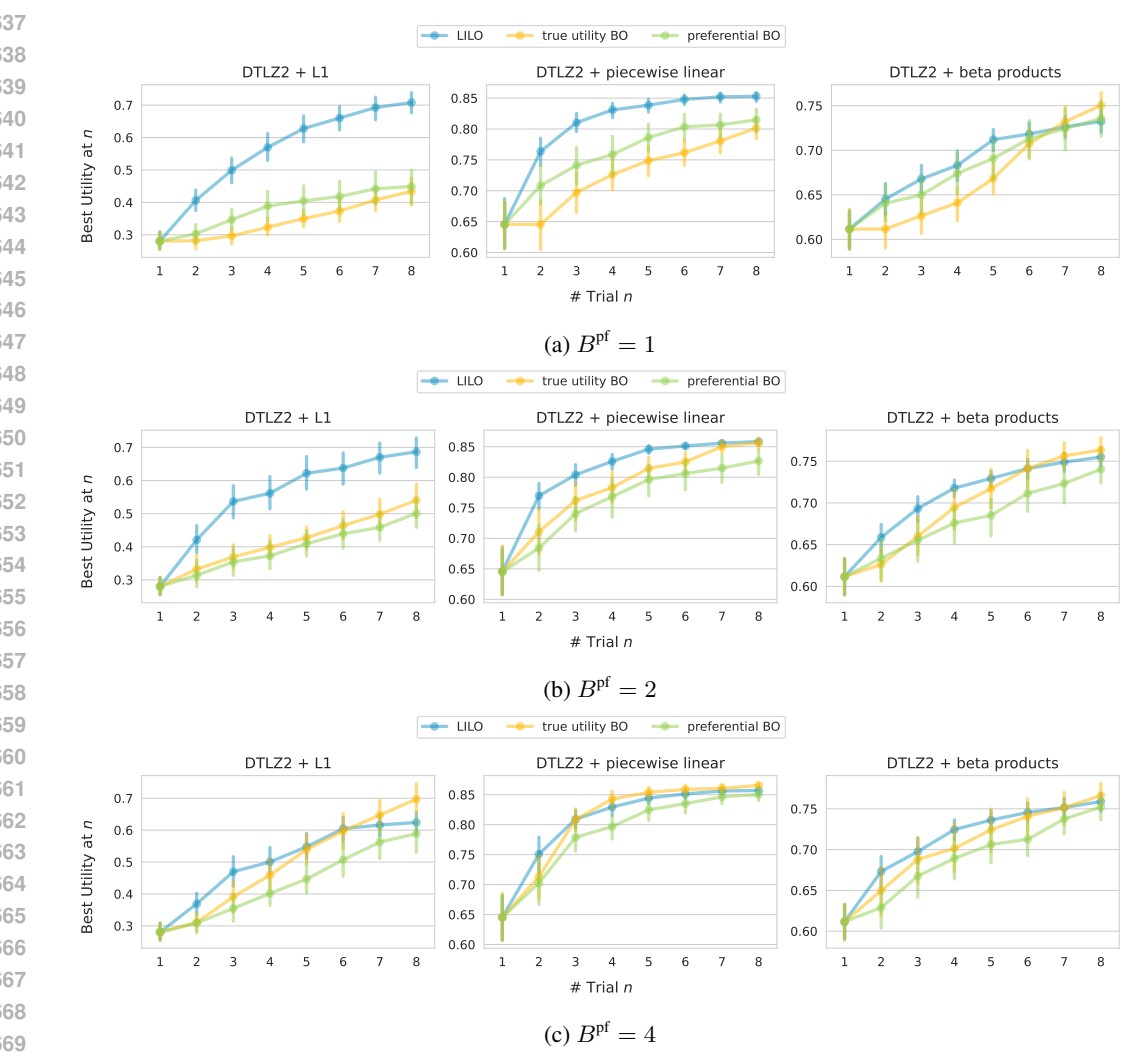

(a) $B^{\text{pf}} = 1$

(b) $B^{\text{pf}} = 2$

(c) $B^{\text{pf}} = 4$

Figure 6: *LILO vs. baselines for different values of $B^{pf}$.* Max value of the ground-truth utility achieved after $n$ iterations. Error bars are 95% confidence intervals of the mean max value across 30 simulation replications.

Table 1: *LLM ablation study.* Max value of the ground-truth utility achieved after $n$ iterations. Error bars are 1 standard deviation of the mean across 30 simulation replications.

(a) DTLZ2 + L1

| method
# trial | LILO
(Llama-3.3-70b) | LILO
(Llama-4-scout) | LILO
(Qwen3-14B) | preferential BO | true utility BO |
|---|---|---|---|---|---|
| 1 | 0.28 ± 0.01 | 0.28 ± 0.01 | 0.28 ± 0.01 | 0.28 ± 0.01 | 0.28 ± 0.01 |
| 2 | 0.42 ± 0.02 | 0.43 ± 0.02 | 0.42 ± 0.02 | 0.31 ± 0.02 | 0.33 ± 0.02 |
| 3 | 0.54 ± 0.03 | 0.53 ± 0.02 | 0.54 ± 0.03 | 0.35 ± 0.02 | 0.37 ± 0.02 |
| 4 | 0.56 ± 0.03 | 0.59 ± 0.02 | 0.59 ± 0.03 | 0.37 ± 0.02 | 0.4 ± 0.02 |
| 5 | 0.62 ± 0.02 | 0.64 ± 0.02 | 0.63 ± 0.03 | 0.41 ± 0.02 | 0.43 ± 0.02 |
| 6 | 0.64 ± 0.02 | 0.67 ± 0.02 | 0.66 ± 0.03 | 0.44 ± 0.02 | 0.46 ± 0.02 |
| 7 | 0.67 ± 0.02 | 0.7 ± 0.02 | 0.69 ± 0.02 | 0.46 ± 0.02 | 0.5 ± 0.02 |
| 8 | 0.69 ± 0.02 | 0.71 ± 0.02 | 0.71 ± 0.02 | 0.5 ± 0.02 | 0.54 ± 0.03 |

(b) DTLZ2 + beta products

| method
# trial | LILO
(Llama-3.3-70b) | LILO
(Llama-4-scout) | LILO
(Qwen3-14B) | preferential BO | true utility BO |
|---|---|---|---|---|---|
| 1 | 0.61 ± 0.01 | 0.61 ± 0.01 | 0.61 ± 0.01 | 0.61 ± 0.01 | 0.61 ± 0.01 |
| 2 | 0.66 ± 0.01 | 0.67 ± 0.01 | 0.66 ± 0.01 | 0.63 ± 0.01 | 0.63 ± 0.01 |
| 3 | 0.69 ± 0.01 | 0.69 ± 0.01 | 0.69 ± 0.01 | 0.66 ± 0.01 | 0.66 ± 0.01 |
| 4 | 0.72 ± 0.01 | 0.71 ± 0.01 | 0.7 ± 0.01 | 0.68 ± 0.01 | 0.69 ± 0.01 |
| 5 | 0.73 ± 0.0 | 0.73 ± 0.01 | 0.71 ± 0.01 | 0.69 ± 0.01 | 0.72 ± 0.01 |
| 6 | 0.74 ± 0.01 | 0.74 ± 0.01 | 0.73 ± 0.01 | 0.71 ± 0.01 | 0.74 ± 0.01 |
| 7 | 0.75 ± 0.01 | 0.75 ± 0.01 | 0.73 ± 0.01 | 0.72 ± 0.01 | 0.76 ± 0.01 |
| 8 | 0.76 ± 0.01 | 0.75 ± 0.0 | 0.73 ± 0.01 | 0.74 ± 0.01 | 0.76 ± 0.01 |

(c) DTLZ2 + piecewise linear

| method
# trial | LILO
(Llama-3.3-70b) | LILO
(Llama-4-scout) | LILO
(Qwen3-14B) | preferential BO | true utility BO |
|---|---|---|---|---|---|
| 1 | 0.65 ± 0.02 | 0.65 ± 0.02 | 0.65 ± 0.02 | 0.65 ± 0.02 | 0.65 ± 0.02 |
| 2 | 0.77 ± 0.01 | 0.76 ± 0.01 | 0.76 ± 0.01 | 0.68 ± 0.02 | 0.71 ± 0.02 |
| 3 | 0.8 ± 0.01 | 0.82 ± 0.01 | 0.82 ± 0.01 | 0.74 ± 0.01 | 0.76 ± 0.02 |
| 4 | 0.83 ± 0.01 | 0.83 ± 0.01 | 0.84 ± 0.01 | 0.77 ± 0.02 | 0.78 ± 0.01 |
| 5 | 0.85 ± 0.0 | 0.84 ± 0.01 | 0.85 ± 0.0 | 0.8 ± 0.01 | 0.81 ± 0.01 |
| 6 | 0.85 ± 0.0 | 0.85 ± 0.01 | 0.85 ± 0.0 | 0.81 ± 0.01 | 0.83 ± 0.01 |
| 7 | 0.86 ± 0.0 | 0.85 ± 0.0 | 0.86 ± 0.0 | 0.81 ± 0.01 | 0.85 ± 0.01 |
| 8 | 0.86 ± 0.0 | 0.86 ± 0.0 | 0.86 ± 0.0 | 0.83 ± 0.01 | 0.86 ± 0.0 |

## C.3 PREFERENCE FEEDBACK BATCH SIZE ABLATION (EXTENDED)

It is not straightforward to compare the DM's workload necessary to answer natural language questions vs. providing e.g. pairwise comparisons. The DM's workload would depend heavily on the specific questions being asked, and in the case of pairwise comparisons, it would depend on the complexity of the utility function and the kind of outcomes being presented. Experiments presented in this subsection are aimed at understanding how many pairwise comparisons or queries to the ground-truth utility are roughly equivalent to a single message of the DM in natural language. In Figures 7, 8, and 9, we compare the performance of LILO with $B^{\text{pf}} \in \{1, 2\}$ with the quantitative baselines (true utility BO and preferential BO) with $B^{\text{pf}} \in \{1, 2, 4, 8, 16\}$. The results demonstrate that even one natural language statement can outperform as many as 8-16 pairwise comparisons or point-wise evaluations of the ground-truth utility. The competitive performance of LILO is most pronounced at the very initial stages of experimentation. This underscores the sample efficiency and information density of natural language feedback, when used effectively.

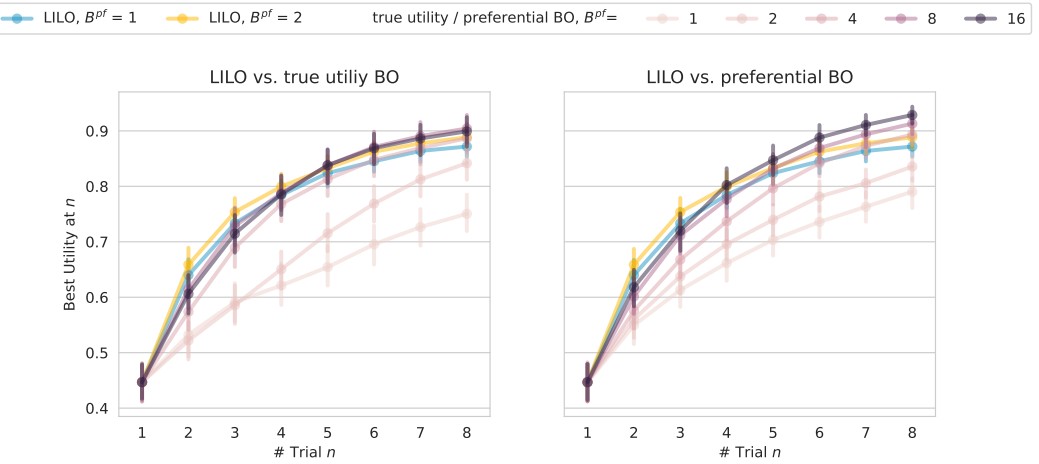

Figure 7: *LILO vs. preferential BO and true utility BO with varying feedback batch size.* Results averaged across all environments, with min-max normalization applied within each environment prior to aggregation.

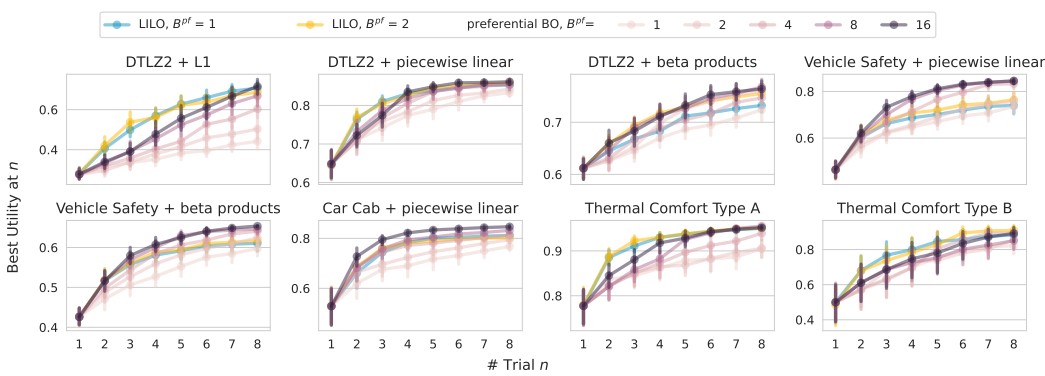

Figure 8: *LILO vs. preferential BO with varying feedback batch size.* Results by environment.

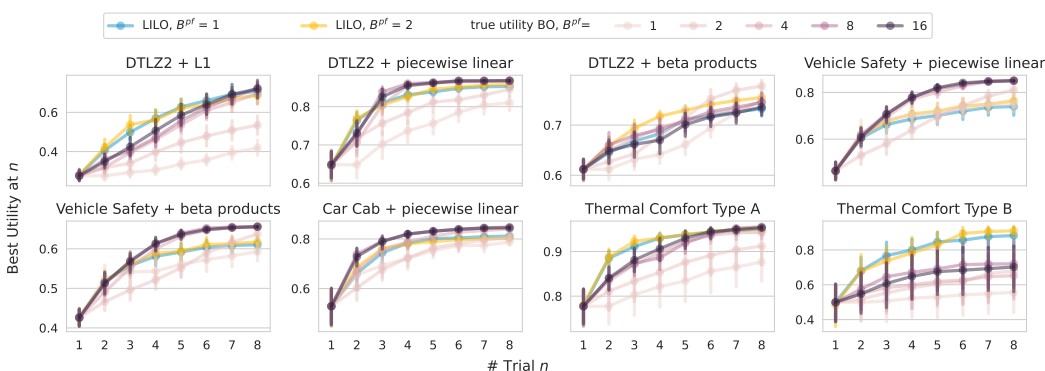

Figure 9: *LILO vs. true utility BO with varying feedback batch size*. Results by environment.

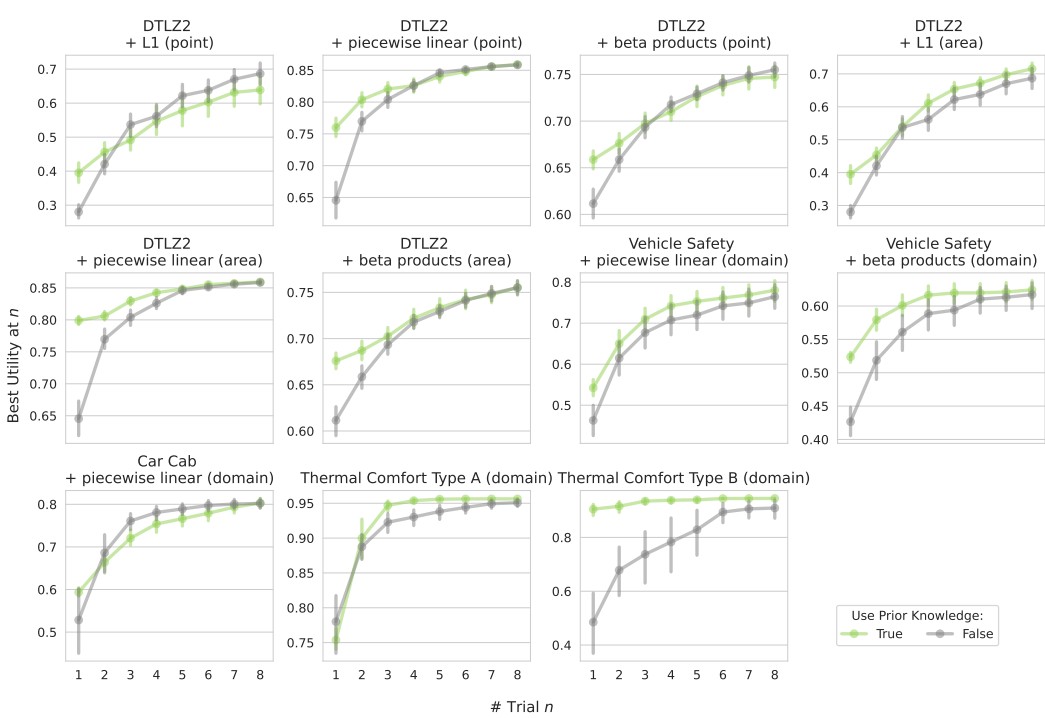

Figure 10: Performance of `LILO` with and without prior knowledge. Results across environments and knowledge types.

## C.4 INCORPORATING PRIOR KNOWLEDGE

In Figure 10 we present an environment-by-environment view of the results from Section 5.3.3.

## C.5 BASELINES – FEEDBACK ACQUISITION ABLATION

This section is aimed to justify the choice of feedback acquisition functions used by the quantitative baselines.

Table 2: *The choice of feedback acquisition function.* Values represent the average of the best ground-truth utility at iteration $n$. Averaged across three simulation environments and 30 replications per environment; min-max standardized within an environment before aggregation. Error bars represent 1 standard deviation of the mean.

(a) True Utility BO

| Method
# trial | EUBO | Max Value Entropy Search | random |
|---|---|---|---|
| 1 | $0.45 \pm 0.017$ | $0.45 \pm 0.017$ | $0.45 \pm 0.017$ |
| 2 | $0.537 \pm 0.017$ | $0.528 \pm 0.017$ | $0.529 \pm 0.017$ |
| 3 | $0.61 \pm 0.017$ | $0.587 \pm 0.017$ | $0.603 \pm 0.017$ |
| 4 | $0.668 \pm 0.017$ | $0.652 \pm 0.017$ | $0.668 \pm 0.017$ |
| 5 | $0.723 \pm 0.017$ | $0.712 \pm 0.017$ | $0.721 \pm 0.017$ |
| 6 | $0.775 \pm 0.016$ | $0.762 \pm 0.017$ | $0.771 \pm 0.016$ |
| 7 | $0.812 \pm 0.016$ | $0.806 \pm 0.017$ | $0.802 \pm 0.016$ |
| 8 | $0.851 \pm 0.015$ | $0.84 \pm 0.016$ | $0.831 \pm 0.016$ |

(b) Preferential BO

| Method
# trial | EUBO | BALD | random |
|---|---|---|---|
| 1 | $0.451 \pm 0.017$ | $0.451 \pm 0.017$ | $0.451 \pm 0.017$ |
| 2 | $0.563 \pm 0.017$ | $0.57 \pm 0.017$ | $0.568 \pm 0.017$ |
| 3 | $0.654 \pm 0.016$ | $0.633 \pm 0.017$ | $0.634 \pm 0.017$ |
| 4 | $0.713 \pm 0.016$ | $0.688 \pm 0.016$ | $0.688 \pm 0.017$ |
| 5 | $0.767 \pm 0.014$ | $0.742 \pm 0.015$ | $0.738 \pm 0.016$ |
| 6 | $0.805 \pm 0.013$ | $0.781 \pm 0.014$ | $0.775 \pm 0.015$ |
| 7 | $0.835 \pm 0.012$ | $0.817 \pm 0.013$ | $0.809 \pm 0.014$ |
| 8 | $0.864 \pm 0.011$ | $0.843 \pm 0.012$ | $0.84 \pm 0.012$ |

For true utility BO, we consider EUBO, Max Value Entropy Search, and a random acquisition function. For preferential BO, replicating the setup of Lin et al. (2022), we compare EUBO, BALD, and a random acquisition function.

**Results.** We compute the results for the three DTLZ2 environments (L1, piecewise linear, beta products). As in the main results, we let $B^{\text{exp}} = d$ and $B^{\text{pf}} = 2$. Table 2 shows the results averaged across all three environments. Overall, the differences in performance are not substantial among alternatives. Across all methods, we observe a slight advantage of EUBO against the alternatives, hence we have chosen it as a default choice of the acquisition function for the baseline methods.

## C.6   LLM'S ACCURACY AND THE FITTED UTILITIES

**LLM's accuracy.**   First, we look at the accuracy of `LILO` in generating pairwise preference labels in step 3. of our algorithm. Figure 11 shows the average accuracy at each trial, computed across 16 seeds and the 3 DTLZ2 environments. We find that already at the first iteration, `LILO` yields high-fidelity predictions, with an average accuracy of 85%. As more information about the optimization objective is gathered, its accuracy reaches values above 90%.

**Qualitative analysis.**   In Figures 13 and 14, we show the plots of fitted utility functions $M_n^y : \mathcal{Y} \to \mathbb{R}$ in comparison to the ground-truth utility function $g : \mathcal{Y} \to \mathbb{R}$. The plots are presented for both `LILO` and the preferential BO baseline. We generate these plots by fixing the center point of the outcome space $\mathcal{Y}$ and varying one of the outcome dimensions in its range. The plots show the posterior means of the GP models $M_n^y$ averaged across 16 replications. We

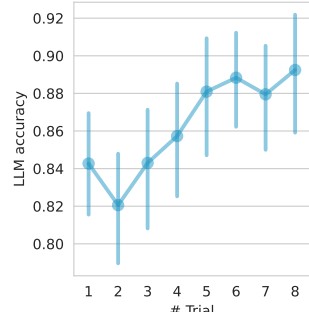

Figure 11: The accuracy of `LILO` in generating pairwise preference choices in step 3. of the BO loop.

observe that with just a few iterations of the BO loop, `LILO`'s fitted utility functions capture the important properties of the ground-truth and qualitatively resemble the shape of the ground-truth much better than the preferential BO baseline. For the piecewise linear utility, the fitted GP's have the required monotonicity properties. Moreover, the relative range of the fitted utilities across the dimensions also matches the relative contributions of each outcome to the overall utility – an increase in the first outcome corresponds to larger utility gains than an increase in the last outcome. In terms of the L1 function, we observe that the peaks of the utility function match well the ground-truth – the utility is supposed to be maximized at a point closest to a desired optimum $y^*$.

## C.7   LONGER TRIALS

We note that this paper primarily concerns settings where configurations are very costly to evaluate, and therefore in practice the number of trials is very limited. Our overarching goal is to minimize human effort during optimization, which is why the main experiments focus on the impact of queries after only a few batched rounds. Nevertheless, in this section, we present additional results evaluating the performance of `LILO` over longer horizons, with up to 16 batched experimental trials. Figure 12 compares `LILO` to both preferential and true utility BO (excluding the two LLM baselines, as they do not show meaningful progression across iterations). We observe that `LILO` maintains competitive performance even in this extended setting. As noted in the main results, the advantage of `LILO` tends to diminish as the number of experimental trials increases, with quantitative baselines eventually catching up and, in some cases, surpassing `LILO` in the long run. This behavior is expected – the in-context learning capabilities of LLMs are inherently limited and do not offer the convergence guarantees provided by conventional baselines operating on quantitative feedback.

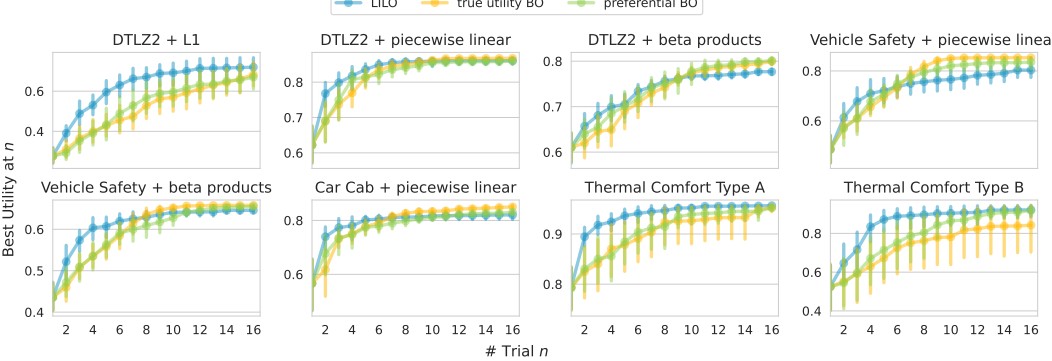

Figure 12: *LILO vs. true utility and preferential BO on longer horizons*. Results aggregated across 16 replications per environment.

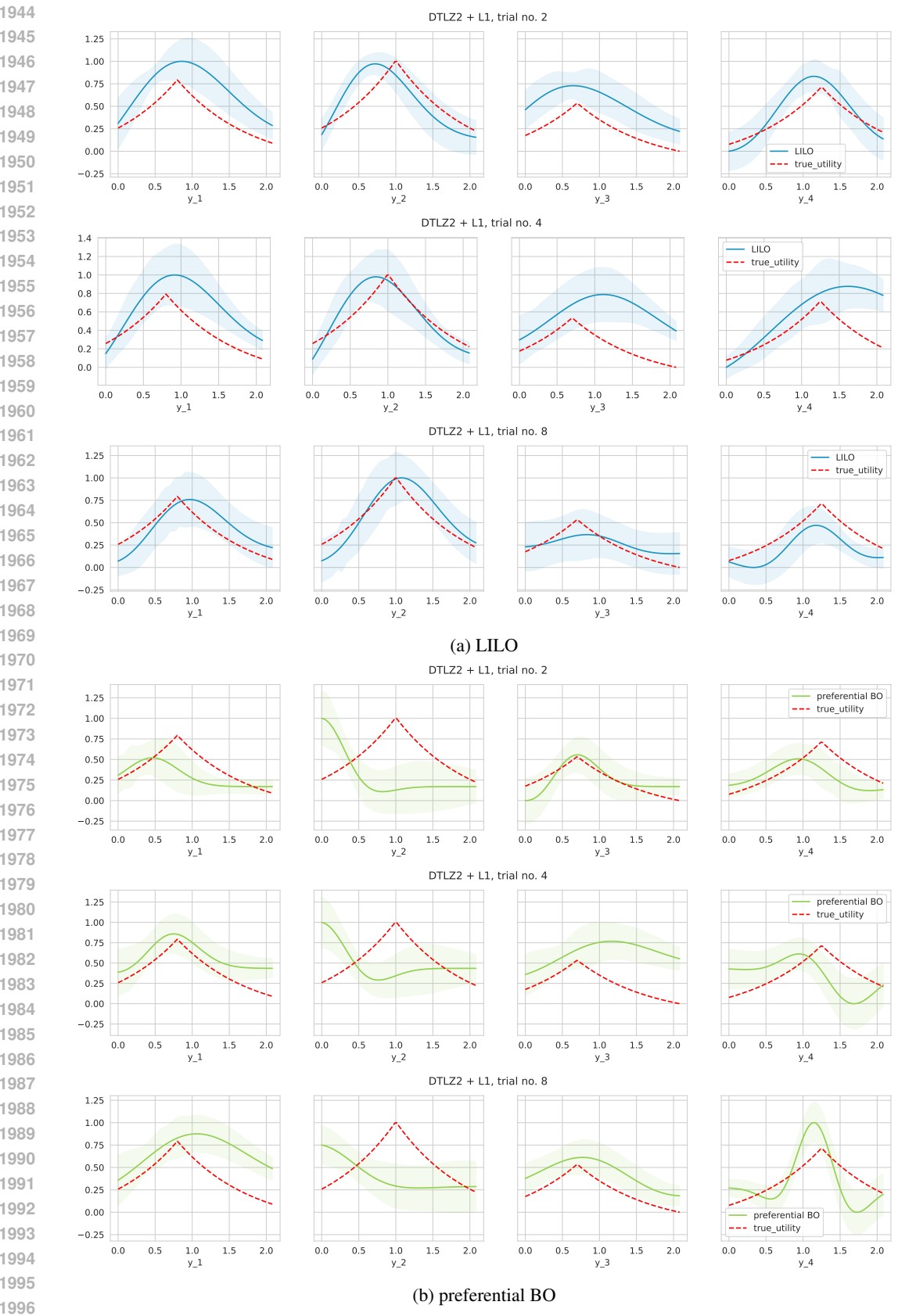

Figure 13: *DTLZ2 + L1: Fitted GP utility models $M_n^y : \mathcal{Y} \to \mathbb{R}$ in comparison to the ground-truth utilities.*

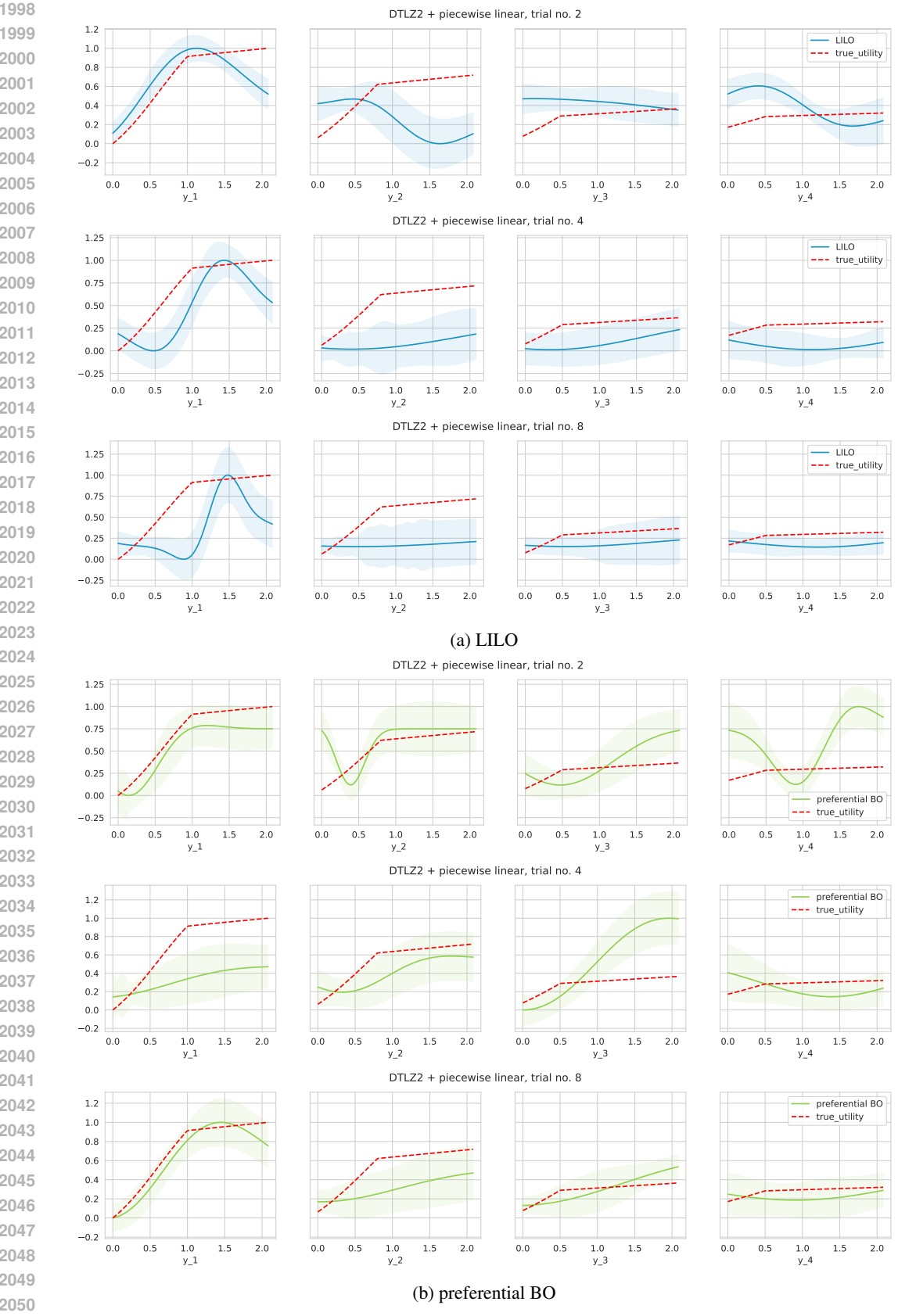

Figure 14: *DTLZ2 + piecewise linear: Fitted GP utility models $M_n^y : \mathcal{Y} \to \mathbb{R}$ in comparison to the ground-truth utilities.*

## C.8   ACQUISITION FUNCTION ABLATION

In our main experiments, we demonstrate the performance of LILO against baselines with all methods using the (Noisy) Expected Improvement as the acquisition function for candidate generation. However, this choice can be replaced with other alternatives. In this section, we demonstrate that LILO maintains competitive performance irrespective of this choice.

In Figures 15 and 16 we demonstrate the performance of LILO and the quantitative baselines with three different choices of acquisition functions for candidate generation: (Noisy) Expected Improvement (as in the main text), Upper Confidence Bound (UCB) with the parameter $\beta$ controlling the trade-off between exploration and exploitation set to 0.5, and Thompson Sampling. All other elements of LILO and the baselines stay fixed. Figure 15 shows results summarized across all environments considered and Figure 16 shows a detailed view of the results. We observe that the Expected Improvement and the UCB acquisition functions, overall, perform better than Thompson Sampling on our selected set of test problems. The advantage of LILO against the baselines is most competitive in these two settings. With Thompson Sampling, LILO performs similarly to the true utility BO baseline, slightly outperforming it at the very first iterations.

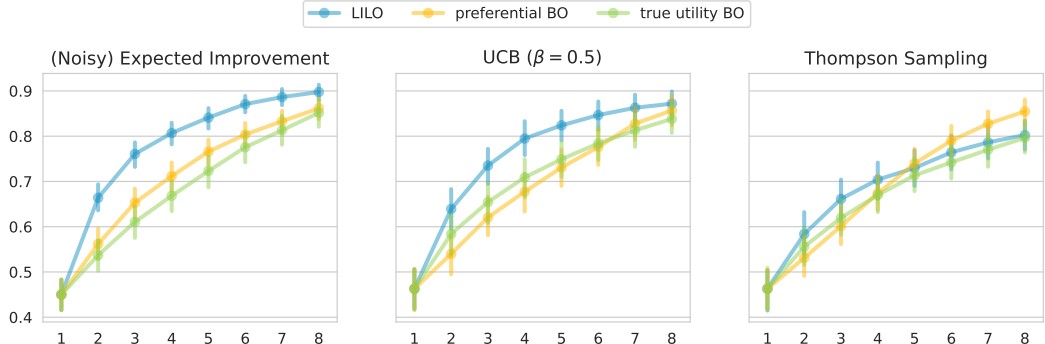

Figure 15: *LILO vs. quantitative baselines using different acquisition functions for candidate generation.* Results averaged across all 8 environments with 16 replications per environment.

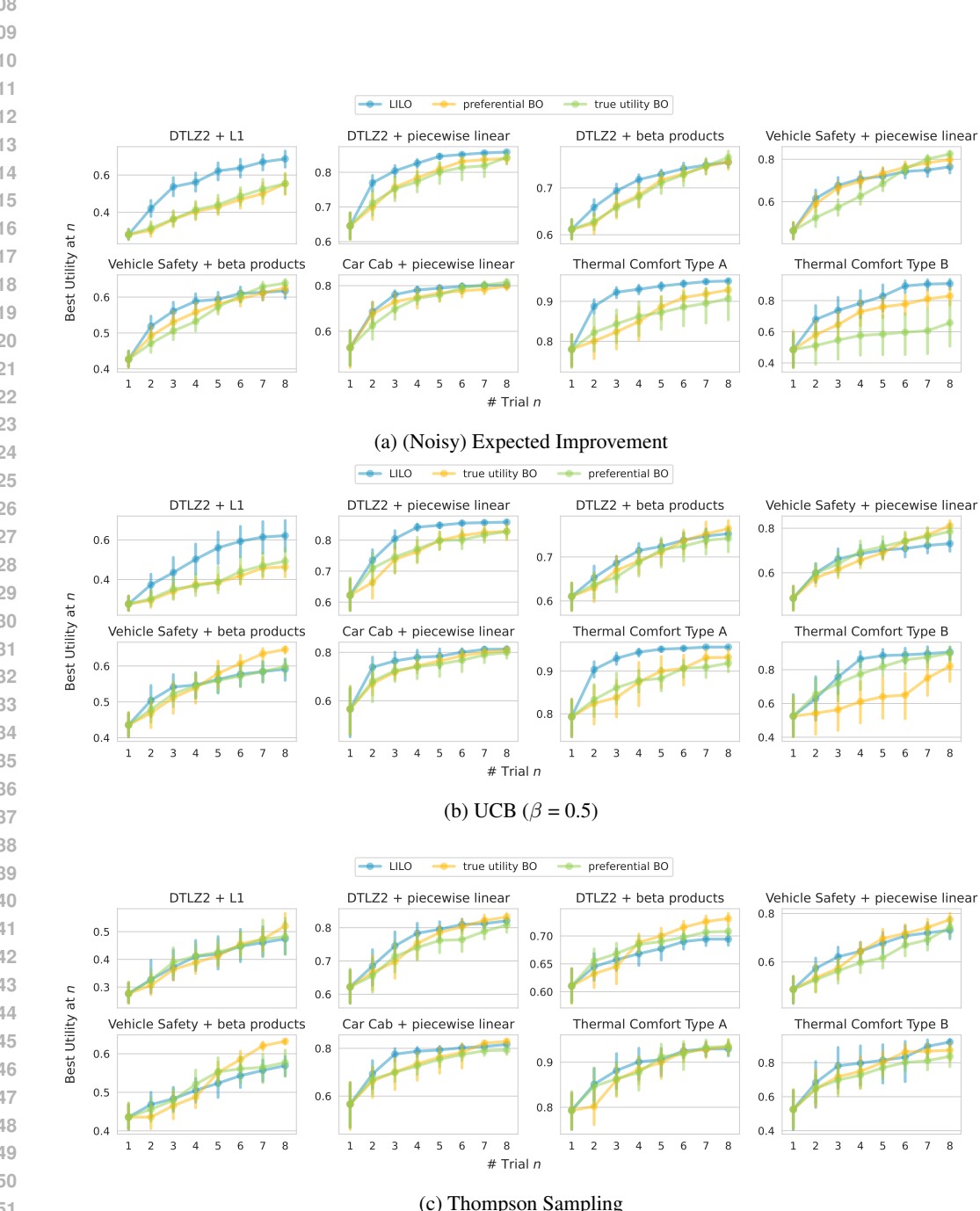

Figure 16: *LILO vs. quantitative baselines using different acquisition functions for candidate generation. Detailed results across all 8 environments with 16 replications per environment.*

## D EXAMPLE CONVERSATIONS FROM THE BENCHMARKS

We present example conversations of LILO with the DM agent to showcase what are the typical questions that our agent asks and what is the form of feedback that it receives. We would like to highlight how for the Thermal Comfort environment the simulated human responses are more qualitative rather than quantitative in nature.

---

**Example conversation (DTLZ2 + piecewise linear)**

```
- DM: My goal is to achieve the following thresholds in each outcome:
y_1 >= 1.0, y_2 >= 0.8, y_3 >= 0.5, y_4 >= 0.5. Improvements over the
thresholds are always good, but less important than bringing the
outcomes to their threshold values. The further away an outcome is
from its threshold, the higher is its negative impact on the overall
performance.
- LILO: How do you prioritize improvements in outcomes that are
already above their threshold values compared to those below their
threshold values?
- DM: I prioritize improvements in outcomes that are below their
threshold values over those that are already above, as meeting these
thresholds has a significant impact on my overall satisfaction.
- LILO: Are there any interactions or trade-offs between outcomes that
 you consider when evaluating overall satisfaction (e.g., is
underperforming in one outcome acceptable if others overperform)?
- DM: Yes, I consider interactions and trade-offs between outcomes,
taking into account their relative importance to my overall
satisfaction, which is not solely determined by their threshold values
 but by the impact of their improvement on my satisfaction.
- LILO: Considering outcomes $1_0$ and $1_2$, how do you trade off
between $y_1$ being slightly above its threshold in $1_0$ and $y_4$
being significantly above its threshold in $1_2$, given that $y_2$ and
 $y_3$ are below their thresholds in both cases? Specifically, which
outcome do you prefer and why?
- DM: I prefer outcome $1_0$ because $y_1$ being slightly above its
threshold of 1.0 takes precedence due to its higher importance.
- LILO: Considering outcomes $2_0$ and $2_1$, both have $y_1$ above
its threshold, but $y_2$ is below its threshold in both cases. Given
that $y_4$ in $2_0$ is above its threshold and in $2_1$ is
significantly below, and $y_3$ is slightly above its threshold in both
, which outcome do you prefer and why?
- DM: I prefer outcome $2_0$ because it has $y_4$ above its threshold,
 contributing positively to my overall satisfaction, unlike $2_1$
where $y_4$ is significantly below its threshold.
```

---

Example 2: Example fragments of conversation of LILO with the DM on the DTLZ2 + piecewise linear environment.

> **Thermo + type A**
>
> – DM: My goal is to keep all metrics within my thermal comfort
> preferences.
> – LILO: Can you provide more specific thresholds for each metric that
> you consider to be within your thermal comfort preferences?
> – DM: To be honest, I prefer it when the air doesn't feel drafty at my
> neck, and I don't feel too much warmth coming from the ceiling. I
> also like it when there's not a big difference in temperature between
> my head and ankles. As for the floor, I find it comfortable when it's
> around 22-24 degrees Celsius. If it's too cold or too hot, my feet
> feel uncomfortable, and that affects my overall comfort.
> – LILO: How do you prioritize these metrics relative to one another?
> For example, is minimizing PPD more important to you than minimizing
> DR, or are they equally important?
> – DM: For me, it's really important that I don't feel drafty or too
> warm from the ceiling. If I had to choose, I'd say minimizing those
> feelings is a bit more important than the other factors. But, at the
> same time, I also value not having a big temperature difference
> between my head and ankles. It's all about finding a balance, I
> suppose. If one thing is off, it can really affect how I feel overall.
> – LILO: Considering arm_index 1_0 (PPD: 25.67%, DR: 13.52%, dT_vert:
> 0.24K, dT_pr: 1.02K, T_floor: 18.47C) and arm_index 1_3 (PPD: 9.02%,
> DR: 3.27%, dT_vert: 4.14K, dT_pr: 17.08K, T_floor: 25.52C), which one
> do you prefer and why?
> – DM: I prefer arm_index 1_3. I feel more comfortable with the
> conditions in 1_3, even though the temperature difference between my
> head and ankles is a bit higher. The air feels calmer and less
> turbulent, which makes me feel more at ease. In 1_0, I feel a bit
> chilly, especially on the floor, which is slightly too cold for my
> liking. I wish it were a bit warmer, around 20-22C would be perfect.
> – LILO: On a scale of 1-5, how satisfied are you with arm_index 1_5 (
> PPD: 57.90%, DR: 0%, dT_vert: 0.50K, dT_pr: 19.41K, T_floor: 27.10C),
> and would you like to see improvements in any specific metrics?
> – DM: I'd rate my satisfaction with arm_index 1_5 as a 1 out of 5. I
> feel quite uncomfortable with these conditions. The air feels stale,
> and I feel a bit too warm from the ceiling, which is making me feel
> uneasy. Also, the floor temperature is a bit too high, I prefer it
> when it's around 20-24C. I'd like to see improvements in reducing the
> warmth from the ceiling and adjusting the floor temperature to a more
> comfortable range.

Example 3: Example fragments of conversation of LILO with the DM on the Thermal Comfort +
Type A environment.

