# OpenReview forum: "LILO: Bayesian Optimization with Interactive Natural Language Feedback"
_ICLR.cc/2026/Conference — Submitted to ICLR 2026_

### Official Review · Reviewer_EgVp · 2025-10-23

**Soundness:** 1
**Presentation:** 1
**Contribution:** 1
**Rating:** 2
**Confidence:** 5

**Summary:**

The paper proposes a natural language preference elicitation method where: 1) two Gaussian processes (GPs) are used for belief maintenance over candidate utilities, 2) a noisy greedy acquisition function ("noisy expected improvement") selects a batch of candidates, 3) an LLM uses the selected batch to generate a set of preference eliciting natural language questions 4) a greedy acquisition function ("expected value of best option") selects (pairs of) candidates to get LLM judgements over. Based on experiments performed with an LLM user simulator and several underlying utility functions, the authors report improvements in utility maximization over LLM-only baselines.

**Strengths:**

- The authors aim to address the important research problems of active LLM-driven preference elicitation (PE), how to balancing exploration and exploitation in natural language PE, and how to combine LLM-based methods with established PE methods such as GP PE bandits.
- The authors perform numerous simulations across several problem domains such as "Vehicle Safety", "Thermal Comfort", and "Car Cab".
- The insight that using LLMs for pairwise comparison improved performance vs pointwise (scalar) judgments is interesting.

**Weaknesses:**

- The motivation for using two GPs (195, 196) for 1) mapping \mathcal{X} to a utility value (a real number) and 2) mapping y = f(x) to a utility value is not at all clear, the large existing volume of work on Bayesian optimization practically always relies on 1) only and the reason why the authors think it is a good idea to deviate from this setting does not come across.
- It is not clear how the two GPs interact if at all for belief maintenance and candidate acquisition.
- The pairwise GP (253) is not defined anywhere, it is just referenced -- the paper should be mostly self contained.
- The candidate selection acquisition function (209), "Noisy Expected Improvement", which appears to just be the noisy greedy acquisition function, is never formally defined. It seems to be adapted to select a batch of candidates as opposed to a conventional acquisition function which selects one candidate, but this adaptation is not discussed. The same comment goes for the EUBO acquisition function (249) for LLM-judge candidates, which appears to be the greedy function adapted for a batch of pairs.
- Only one acquisition function per candidate selection step (one for question generation candidates (209), one for LLM-judge candidates (249)) is tested and discussed -- there are many alternatives such as UCB, entropy reduction, and Thomson sampling which are extremely common.
- Many similarities to [1] which is never referenced.

[1] Handa, Kunal, et al. "Bayesian preference elicitation with language models." arXiv preprint arXiv:2403.05534 (2024).

**Questions:**

- The "black-box" function in standard Bayesian Optimization for PE is the user's utility over the candidate set "\mathcal{X}". Why is there a need to introduce another function f(x) and model it with a second GP?
- What are the formal definitions of a pairwise GP (including belief updates) and the acquisition functions?
- Were other acquisition functions could be explored, for both candidate selection for question generation and for selecting candidates for LLM evaluation?

---

> ### Author Response · Authors · 2025-11-22
>
> Thank you for your feedback. We appreciate the reviewer’s concerns which we address below point by point.
>
> ---
>
> ## Weaknesses
>
> **W1 & W2. The setup and the use of two GPs.**
>
> Our setup involves a composition of two functions: the outcome function $y = f(x)$ and the utility function $u = g(y) = f(g(x))$. This setting precisely aligns with the framework employed by Astudillo & Frazier (2020) and Lin et al. (2022), as described in the paper. This formalism is applicable to setups in which hyperparameter $x \in \mathcal{X} \subset \mathbb{R}^d$ are corrspond to a vector of *multiple* outcomes $y \in \mathcal{Y} \subset \mathbb{R}^k$, over which a utility function can be applied to summarise the decision maker's preferences into a single scalar value $u = g(y) \in \mathbb{R}$. This constitutes a common and a generalized setup across various problem domains, including online experimentation, engineering design, and robotics, where the black-box function yields a vector of outcomes and the decision maker has certain preference over this outcome vector (e.g., the DM prefers the designed vehicle to have higher safety features and less concerned about the weight of the vehicle). This setup is standard in many industry settings, such as Bayesian optimization via A/B tests, scheduling simulations where different configurations induce various many-objective tradeoffs for which decision makers are not able to specify a scalarization in advance.s. We will add a comment to the paper to make this more clear.
>
> We would also like to clarify the use of the two GP models: $M^x: \mathcal{X} \rightarrow \mathbb{R}$ and $M^y: \mathcal{Y} \rightarrow \mathbb{R}$. The $M^x$ model is the key proxy model approximating the composite mapping $g \circ f$ — it is necessary to generate new candidates $x \in \mathcal{X}$ for expermentation based on their expected expected utility and uncertainty. The $M^y$ model plays a supporting role — in LILO, it is solely used for selecting the set of $K$paired outcomes $(y_k, y_k')$ for LLM labelling. As we demonstrate in section 4.3.2, the use of the $M^y$ model for pair selection with respect to EUBO improves the predictive performance of the outcome-space proxy in identifying the best candidate. This is important as, given a large set of already run experiments $D^{\text{exp}} = \{(x_i, y_i)\}$, the $M^y$ model can be used to surface the most promising candidates. It is also expected that in many scenarios, the $g$ function is of far lower complexity than the black-box mapping $f$. Hence the  model approximating $g$ is expected to perform beter in this task than the $M^x$ model approximating $g \circ h.$ At the same time, we do acknowledge that the use of the $M^y$ models does not directly translate to significant improvements in terms of the overall optimization performance, measured with respect to the ground-truth utility function. Thus, in practice, fitting of $M^y$ can be optional.
>
> **W3. Defining the pairwise GP.**
>
> Thank you for asking the questions regarding the procedure of fitting the pairwise GP model. Given a set of observed outcomes $(y_k, y_k')$and binary preference labels $p_k \in \{0, 1\}$, the latent utility function ($g(\cdot)$ is modeled as a GP. The probability that $y_k$ is preferred over $y_k'$ is given by \($\Phi(g(y_k) - g(y_k'))$, where $\Phi(\cdot)$ is the cumulative distribution function of the standard normal distribution. Since the probit likelihood is non-Gaussian, we use the Laplace approximation to perform inference.
>
> The pairwise GP model as described above is a standard model in preference learning and the original paper is widely cited, hence we omitted the full details of this method in our paper. However, given your comment, we have now included a brief description of this procedure in Appendix A.3. and we refer the readers to the original paper for a more comprehensive details.
>
> **W4. Noisy Expected Improvement and EUBO.**
>
> We respectfully clarify that our acquisition function is *not* a noisy greedy strategy. We use qLogNoisyExpectedImprovement from BoTorch (`botorch.acquisition.logei.qLogNoisyExpectedImprovement`), which jointly evaluates the utility of the entire batch via Monte Carlo sampling—rather than selecting points greedily. This implementation follows the work of Ament et al. (2023).
>
> Regarding the EUBO acquisition function for pair selection, the reviewer is correct — we use a greedy approach for selecting a batch of K pairs. For a pair of outcomes $(y_k, y_k')$ EUBO has an explicit analytic form. We thus employ the AnalyticExpectedUtilityOfBestOption from BoTorch (`botorch.acquisition.preference.AnalyticExpectedUtilityOfBestOption`) to compute its value for all possible combination of pairs and select the set of $K$ pairs with the highest utilties.
>
> To improve the clarity of presentation, we have updated our submission to explicitly reference these methods. Thank you for bringing our attention to this matter.

---

> > ### Author Response · Authors · 2025-11-22
> >
> > **W5. Other acquisition functions.**
> >
> > Although various acquisition functions exist and are applicable, we emphasize that the primary contribution of this paper is not the acquisition function itself, but rather the synergistic integration of the LLM within the BO framework. Regarding the choice of the acquisition function for candidate generation, we highlight that both the preferential BO and true utility BO use the same acquisition function — LogNEI, which makes our comparisons fair.
> >
> > Regarding the choice of EUBO for paired outcome selection, during development of this work, we did conduct experiments utilizing BALD (Bayesian Actively Learning by Disagreement) and found that EUBO, which focuses on high-utility regions, yielded marginally better performance in collecting pairwise comparison data from the LLM. Consequently, we chose to use EUBO for selecting and labeling data points with the LLM in LILO. This choice aligns with the established practice in the BOPE paper (Lin et al. 2022) and is consistent with what’s used in the preferential BO baseline (see Appendix C.5. for an ablation of this choice).
> >
> > **W6. LILO vs. Handa et al. [1]**
> >
> > Thank you for pointing us to the related work of Handa et al. [1]. While closely related in spirit, their setup is different from ours. Similar to the PEBOL framework of Austin et al. (2024), Handa et al. assume a finite, discrete set of candidate options, each represented through known binary feature vectors, and the LLM is used to elicit preferences directly over this discrete candidate *set*. Both works maintain an explicit parametric posterior over the utility of each candidate.
> >
> > In contrast, our method is particularly suited for continuous search spaces, where the underlying candidate domain cannot be enumerated or represented by a fixed set of predefined features. Moreover, our setting assumes a black-box mapping from a continuous candidate vector $x$ to an observable  outcome vector $y$, and the decision-maker provides preferences over outcomes, not over the (latent) candidate representations themselves. We do agree, however, that the work of [1] is related to ours and we have now cited it in the related works section accordingly. Thank you for this suggestion.
> >
> > ### Questions
> >
> > **Q1.** See our response to W1 & W2.
> >
> > **Q2.** See our response to W3.
> >
> > **Q3.** See our response to W5.
> >
> > ---
> >
> > We hope that the clarifications here help reflect the contribution more accurately and justify an upward revision.
> >
> > **References:**
> >
> > - Astudillo, R., Frazier P. Multi-attribute Bayesian optimization with interactive preference learning. AISTATS 2020.
> > - Lin, Z. J., Astudillo, R., Frazier, P., & Bakshy, E. Preference exploration for efficient bayesian optimization with multiple outcomes. AISTATS 2022.
> > - Chu, W., & Ghahramani, Z.Preference learning with Gaussian processes. ICML 2005.
> > - Ament, S. et al. Unexpected Improvements to Expected Improvement for Bayesian Optimization. NeurIPS 2023.
> > - Austin, D. et al. Bayesian Optimization with LLM-Based Acquisition Functions for
> > Natural Language Preference Elicitation. RecSys 2024.

---

> ### Comment · Reviewer_EgVp · 2025-11-27
> **Reviewer response to rebuttal**
>
> Thank you for your detailed response and for addressing my feedback -- I've bumped up the soundness and contribution scores and slightly revised my review.
>
> W1/W2: This clarification is helpful. However, if the use of the second (optional) GP "does not directly translate to significant improvements", I find it hard to see why it is part of the proposed method.
>
> W3: I would suggest including all key definitions in the main body, not the Appendix -- I think the pairwise GP could be defined in just a few lines -- but this is not a major point.
>
> W5: It is good that the acquisition function is consistent across methods, but I think that it would be important to add results showing whether the results generalize across at least a few variations in the (two-level) acquisition function (the current proposal seems like a highly specific choice to me).

---

> ### Author Response · Authors · 2025-11-28
>
> Dear Reviewer,
>
> Thank you again for the constructive discussion. We are glad to hear that the clarifications were helpful, and we appreciate your updated scores. Below we address your remaining points in turn.
>
> **W1/W2.** We understand the reviewer’s concern regarding the motivation for the supporting  $M^y$ model. As mentioned earlier, the $M^y$ model can be useful for identifying high-yielding outcomes within $D^{\text{exp}}$, which can be then surfaced to the DM for inspection. Our intuition is that this task can be performed more accurately using $M^y$ rather than $M^x$: modeling $g$ directly is typically easier than modeling the composite $g \circ f$, where $f$ is a complex black-box function.
>
> To verify this empirically, we computed the Spearman rank correlation between (i) the posterior mean of $M^y$ and the ground-truth utilities and (ii) the posterior mean of $M^x$ and the ground-truth utilities on the experimental dataset at the final iteration. We find that $M^y$ achieves, on average, a **0.11 higher** Spearman correlation. Concretely, across iterations, $M^y$'s average correlations range in 0.52–0.74, whereas $M^x$'s average correlations range in 0.40–0.65. This supports our intuition that $M^y$ provides a more accurate ranking of outcomes, an thus we believe it remains valuable to maintain this model at a relatively low additional cost.
>
> **W3.** We thank the reviewer for the suggestion to include definitions of the Pairwise GP and EUBO acquisition function in the main text. We agree that these concepts are less standard than classical GPs or common acquisition functions (e.g., EI or UCB). We have therefore added concise explanations in a new Background section of the updated PDF.
>
> **W5.** Following your suggestion, we have extended our ablation studies on acquisition choices. In addition to the existing ablation studies on the feedback acquisition, we have evaluated LILO and all quantitative baselines using **(Noisy) Expected Improvement**, **UCB**, and **Thompson Sampling** for candidate generation. Results are provided in Appendix C.8 of the updated PDF. In summary, LILO exhibits competitive performance across all candidate acquisition variants, indicating that our results generalize beyond the specific choices used in the main experiments.
>
> We hope that these clarifications, along with the additional experimental evidence, resolve your remaining concerns and support a higher overall assessment of our work.

---

### Official Review · Reviewer_B7nh · 2025-10-29

**Soundness:** 2
**Presentation:** 2
**Contribution:** 3
**Rating:** 6
**Confidence:** 4

**Summary:**

In the context of preferential Bayesian optimization (PBO), the paper proposes an algorithm that incorporates an LLM to improve optimization performance. From the main paper (e.g. Algorithm 1), it is clear that the LLM acts an acquisition function in the PBO loop and an assistant of communicating justifications of DM’s preferential feedback, but also as an auxiliary preferential feedback oracle giving directly the DM’s expected choice probabilities (Algorithm 3).  The paper demonstrates the improved performance and sample efficiency compared to various baselines on a variety of semi-synthetic environments.

**Strengths:**

The paper tackles an interesting and timely problem of incorporating large language model capabilities into the preferential Bayesian optimization loop. The paper provides a comprehensive solution to the problem involving end-to-end interaction loop with DM (domain expert) and the experimentation black-box, where the LLM acts in various roles, most importantly processing textual justification data about the DM’s preferences. The experimental section examines various baselines and setups, demonstrating improvements over both pure PBO methods and pure LLM-based baselines.

**Weaknesses:**

I did not find any discussion of potential limitations of using LLMs to directly provide probabilistic judgments, that is the choice probability p_k (e.g. see Prompt 4). In the context of confidence elicitation in LLMs, there is important related work studying this aspect (Xiong et al., 2024). It seems that the proposed approach adopts so-called “black-box method” in contrast to “white-box method” (Xiong et al., 2024). Kadavath et al. (2022) proposed using token-level log-probabilities to estimate the LLM probabilities, i.e. relying on white-box access to internal LLM information (Xiong et al., 2024). A natural question arises: Why the paper adopted black-box method, while Llama is an open-source model, and more calibrated white-box method could have been adopted (Xiong et al., 2024)?
“In this paper, we introduce Language-in-the-loop Optimization (LILO), a framework designed to combine the complementary strengths of BO and LLMs while avoiding their respective weaknesses”.. I think this claim is only partly true. For example, directly prompting an LLM to provide a probability estimate is not one of its strengths (e.g. Kapoor et al., 2024).

The proposed method is an algorithmic patchwork, and the presentation lacks consistency and a well-balanced level of abstraction. As a concrete example, let us consider an apparent inconsistency in the main paper:

Lines 250-252: “For each pair (y_k,y_k’), the LLM determines which outcome is more aligned with the DM’s preference, producing a label p_k \in {\0,1\}  indicating whether or not y_k is preferred over y_k’”.
Lines 163-184 (Algorithm 1): There is no mention LLM making preference judgment nor any mention of data p_k. There is only mention of DM.get_answers, which refers to the decision maker (DM).

However, a deeper look into the appendices reveals that the main component of the method (i.e. as the abstract states: “a large language model (LLM) to convert unstructured feedback in the form of natural language into scalar utilities”) is found in Appendix Algorithm 3 “fit proxy models” where it is clear that LILO.get_pairwise_pref (i.e. LLM) gives directly the probability p_k by prompting.
Then, by digging further one can find that in Appendix B.3 “LLM-based simulation of the human preference feedback” there is discussion on using LLM as a DM proxy in the experiments. Still, some important details to evaluate the validity of the empirical experiments are buried in the code (Lines 1277-1279): “For the exact utility-specific versions of this prompt we refer the readers to our code which is made available as part of this submission in the supplementary materials.”

In summary, the manuscript would greatly benefit from revision that places more focus on the claimed main contributions such as “We show how to translate such natural language feedback into quantitative latent utilities that can be used effectively by a surrogate and acquisition function, systematically exploring the design choices required to render this approach both effective and practical.”

References

Kadavath, S., Conerly, T., Askell, A., Henighan, T., Drain, D., Perez, E., ... & Kaplan, J. (2022). Language models (mostly) know what they know. arXiv preprint arXiv:2207.05221.

Kapoor, S., Gruver, N., Roberts, M., Collins, K., Pal, A., Bhatt, U., ... & Wilson, A. G. (2024). Large language models must be taught to know what they don’t know. Advances in Neural Information Processing Systems, 37, 85932-85972.

Xiong, M., Hu, Z., Lu, X., LI, Y., Fu, J., He, J., & Hooi, B. Can LLMs Express Their Uncertainty? An Empirical Evaluation of Confidence Elicitation in LLMs. In The Twelfth International Conference on Learning Representations.

**Questions:**

For preferential BO community, an interesting question is how LLM’s estimates about DM’s choice probabilities used to fit the GP surrogate for the utility function. In the last lines of Algorithm 3, I can find lines “fit_pairwise_gp”, but do not find in the paper what this subroutine does. Do you treat p_k as a scalar value and just GP regression, or do you use it in place of preference likelihood or what?

---

> ### Author Response · Authors · 2025-11-22
>
> We thank the reviewer for their encouraging feedback and comments. We would like to take this opportunity to clarify the details of our method and address the reviewer’s concerns.
>
> ---
>
> **Probabilistic judgements.** We would like to clarify that in the default version of LILO (with pairwise preference choices, not LILO (scalar)), the LLM is not making any probabilistic judgements. The preference labels p_k are binary, indicating the preference choice between two alternative outcomes $y_k$ and $y_k’$ given the DM’s feedback. This is stated, as the reviewer quotes, in lines 250-252: “For each pair $(y_k,y_k’)$, the LLM determines which outcome is more aligned with the DM’s preference, producing a label $p_k \in \{0,1\}$ indicating whether or not $y_k$ is preferred over $y_k'$. Note, the set notation for $p_k$ taking only values 0 or 1. Prompt 3 is used for eliciting these binary pairwise comparisons. The uncertainty estimates are then obtained via the pairwise GP model.
>
> **White-box vs. black-box access to LLMs.** We appreciate the reviewer’s suggestion regarding white-box methods, such as using token-level log-probabilities for confidence elicitation. While we indeed work with open-source models in this paper, our primary goal was to design a method that is broadly applicable to both open- and closed-source LLMs.
>
> **LILO vs. LILO (scalar).** During the development of our method, we also tried prompting LILO to provide scalar estimates in range [0, 1] (i.e. pseudo probabilities) that the DM would be satisfied with a given experimental outcome y, given their feedback. However, as demonstrated in Figure 4 (a), this approach led to an inferior performance. As the reviewer mentions, providing probability estimates directly is not a strength of LLMs, and our results conform with this observation.
>
> **New insights.** Thank you for this helpful suggestion. In addition to the existing studies (LILO vs. LILO (pairwise) in Section 4.3.1 and the comparison of pairwise-labelling methods in Section 4.3.2), we have expanded our evaluation to place greater emphasis on our core contribution—namely, providing a method for translating natural-language feedback into quantitative latent utilities suitable for BO.
>
> In direct response to the reviewer’s request for clearer evidence supporting this contribution, we have added a new set of analyses in Appendix C.6. These results show that our approach can recover the ground-truth utilities with high fidelity, both quantitatively (via accurate pairwise labelling) and qualitatively (via the shape of the fitted utility functions). The additional evidence directly supports the contribution highlighted by the reviewer and further clarifies why the proposed design choices are effective.
>
> **Presentation of LILO.** Thank you for raising the concern about the details of the pairwise preference labelling subroutine being not clearly presented. We would like to firstly highlight that this part of our algorithm is explained in words in the paragraph “Utility Estimation via LLM Labeling + GP Modeling”. We also explicitly point out this part of the algorithm in the main illustration of LILO on Figure 2. Regarding Algorithm 1, due to space constraints, the subroutine of pairwise preference labelling and GP fitting is collapsed inside the fit_proxy_models method which is then expanded in Algorithm 3 of the Appendix. To make this clear we have now included a direct reference to Algorithm 3 in the updated version of our manuscript. We are also open to considering moving Algorithm 3 to the main text, if the reviewer finds it beneficial for an improved readability. Thank you for helping us improve the presentation of our method.
>
> **Fitting pairwise GP.**  Thank you for asking the questions regarding the procedure of fitting the pairwise GP model. As a reminder, the collected dataset for fitting the model is of the form ${(y_k, y_k’, p_k)}$ where $y_k$ and $y_k’$ are two outcomes and $p_k$ a binary label indicating the preference choice. To fit a pairwise GP, we used the probit-likelihood GP with Laplace approximation as introduced by Chu & Ghahramani (2005) for the pairwise GP model. This is a standard model in preference learning and the original paper is widely cited, hence we omitted the full details of this method in our paper. However, given your comment, we have now included a brief description of this procedure in Appendix A.3. and we refer the reader to the original paper for more comprehensive details.
>
> ---
>
> We hope the additional evidence and clarifications help address the reviewer’s reservations and support a more favourable assessment of our work. If the reviewer has any outstanding questions or concerns we are keen to engage in a further discussion.
>
>
> **References:**
> - Chu & Ghahramani, Preference Learning with Gaussian Processes. ICML 2022.

---

### Official Review · Reviewer_BnvK · 2025-11-03

**Soundness:** 2
**Presentation:** 3
**Contribution:** 2
**Rating:** 4
**Confidence:** 3

**Summary:**

This paper proposes Language-in-the-loop Optimization (LILO), a novel framework which integrates Bayesian Optimization (BO) with natural language feedback from a human decision-maker (DM). The key idea is to use a LLM as a translator component within a standard BO loop. LILO allows the DM to provide free-form text feedback, and then a LLM then interprets this text feedback to generate pairwise preference labels for observed outcomes. The LLM-generated labels are used to train a Gaussian Process (GP) preference model, which in turn guides the BO acquisition function to select new candidates. Experiments on several tasks (DTLZ2, Vehicle Safety, etc.) show that the proposed method outperforms both LLM-only optimizers (which lack principled uncertainty quantification) and traditional BO methods.

**Strengths:**

The paper identifies a limitation in standard and preferential BO. Using natural language rather single scalar values is intuitive to formulate complex real-work objectives. The study is well-motivated.

The design of using a LLM in the BO framework is reasonable, and the empirical finding is promising. Also, the ablation study provides useful observations, such as pairwise comparisons provide more reliable utility estimates than direct scalar predictions.

**Weaknesses:**

The experimental results are from simulated environments, which does not reflect the real-world challenges and makes the strong claims not convincing.

In each iteration, true utility BO receives $B^{pf} = 2$ scalar outcomes, while LILO receives $2$ text natural language answers. As also argued by the authors, the text feedback is much richer. Therefore, the comparison seems not fair. A fairer comparison would be against a preferential BO baseline given a larger budget of pairwise comparisons, or a true utility BO with larger $B^{pf}$.

**Questions:**

Please see the weakness section.

---

> ### Author Response · Authors · 2025-11-22
>
> Thank you for your feedback and comments which we address below:
>
> **W1. Simulated environments**
>
> Our experimental setups are motivated by real-world applications and are consistent with the practices adopted in recent publications in the field. We appreciate your concern regarding the realism of our simulations. If you could kindly specify which aspects of the setups you find less representative of real-world challenges, we would be grateful for the opportunity to address your concerns in more detail and further improve our work.
>
> **W2. Fairness of comparison**
>
> Thank you for raising this point. We refer the reviewer to the paragraph *Fairness of comparison* of the main rebuttal  and the newly added Appendix C.3. for a detailed discussion of this topic including new experimental results of LILO compared to baselines with larger batch sizes.
>
> ---
>
> We hope our answer addresses the reviewers concerns. If there are any outstanding questions, we are eager to engage in a further discussion and provide follow-up clarifications.

---

### Official Review · Reviewer_Q6Ph · 2025-11-04

**Soundness:** 2
**Presentation:** 2
**Contribution:** 2
**Rating:** 2
**Confidence:** 4

**Summary:**

The paper proposes LILO, a BO pipeline that: (i) converts natural-language feedback into pairwise preferences using an LLM, (ii) fits two GP surrogates—$M_y:Y\rightarrow U$ from labeled outcome pairs and $M_x: X\rightarrow U$ for candidate selection—and (iii) uses LogNEI to pick new experiments and EUBO to choose outcome pairs for labeling; an optional “language prior” can warm-start round 1.

**Strengths:**

The paper is easy to follow and the writing is clear. The proposed method is easy to reproduce and analyze.

**Weaknesses:**

1. Evaluation is dated and narrow. Core results are on four legacy simulators: DTLZ2 (2002), Vehicle Safety, Car Cab Design (2008), and Thermal Comfort (1970/2005). There’s no modern HPO or realistic high-dimensional outcome task (e.g., image/text) despite the paper’s motivation. The paper even acknowledges applicability to high-dimensional outcomes without demonstrating it.

2. the DM prompt contains ground-truth utilities $g(y_i)$ or observed outcomes—an unrealistic advantage for a human and a potential leakage channel shaping the language. e.g., the paper says line 342-344, "For methods involving natural language feedback, answers to questions posed by the LLM agent are simulated with another LLM containing a textual description of the ground-truth utility function
in the prompt ". And in prompt 7, The prompt tells the DM:“You have observed the following outcomes with their corresponding utility values and contributions to the overall utility.”

3. Short horizon and heavy feedback budget: The loop runs only T=8 rounds with
$B_{exp}=d$ and $B_{pf}=2$, while internally labeling K=64 pairs per round—an unusually large annotation budget relative to very few optimization steps, which may not reflect real-world costs.

4. Missing baselines: The paper compares to oracle true-utility BO, preferential BO, and two LLM-only variants (one “LLAMBO-like”). That’s a fair start, but it omits competitive BO+LLM or preference-BO variants such as embedding-based surrogates over language/strings [1], best-of-k preference-BO (top-k ranking) [2], and multi-objective preference-BO with learned/implicit scalarizations [3]—each of which would stress whether LILO’s dual-GP design is really needed.

5. The paper lacks novelty: the optimization math (GP surrogates, preference likelihood, EUBO/NEI acquisitions) is standard BO/PBO; the main contribution is a well-engineered placement of an LLM to turn text into pairwise labels.

---

[1] Nguyen et al., Language Model Embeddings Can Be Sufficient for Bayesian Optimization, 2024.

[2] Nguyen et al., Top-k Ranking Bayesian Optimization, AAAI 2021.

[3] Ozaki et al., Preferential Multi-Objective Bayesian Optimization, 2024.

**Questions:**

What safeguards ensure the DM’s responses don’t implicitly encode the exact utility (given that the DM was shown g(y) and that LILO doesn’t overfit to such patterns?

---

> ### Author Response · Authors · 2025-11-22
>
> We thank the reviewer for their time and constructive feedback, which helped us improve our work. In our response, we have expanded the appendix to include new results and we address the reviewer’s comments below point by point.
>
> ---
>
> ### Weaknesses
>
> **W1. Evaluation.**
>
> We would like to note that most HPO benchmarks available focus on only a few if not just one outcome (often being a certain measure of accuracy), rendering little room for designing non-trivial utility functions with tradeoffs between outcomes to be considered.
>
> Regarding high-dimensional outputs, in general, LILO is directly applicable to high-dimensional outcome problems by using the $M^x$ model only (skipping the model $M^y$). Notably, in Figure 4(b), we show that LILO relying purely on $M^x$ performs competitively compared to other variants of LILO. While evaluating using LILO with very high-dimensional outcomes (texts or images) would be interesting, this is challenging to evaluate across sufficiently many replications due to very high evaluation costs (e.g., the time and cost of rendering multiple images in each trial). Reliably learning complex preferences over very high dimensional outputs would likely require more evaluations than feasible in our setting.
>
> **W2. Prompt for human feedback simulation.**
>
> Thank you for raising this important point. We would like to clarify that the function $g$ represents the internal utility function of the decision maker. In our human feedback simulator, providing information about $g$ is necessary to ensure that the LLM’s responses are well-aligned with the intended preferences. However, we fully agree that care must be taken to prevent the feedback from unrealistically revealing the exact functional form of $g$.
>
> To address this, we adopted a prompt engineering approach to guardrailing. During prompt design, we iteratively refined the instructions to explicitly prohibit the LLM human simulator from revealing the utility function’s formula or referring to its existence. Our final instructions include the following statement:
>
> > *In your final answers, you cannot reveal the explicit formula of the utility function.
> The form and the values of the utility functions is a latent feature of the human expert, thus you should not refer to it explicitly or even mention its existence.*
> >
>
> For specific utility functions, we added further guardrails. For instance, with the beta products utility, we also have the following:
>
> > Because you are emulating a human, you should not describe the functional form of the utility function. E.g., - you cannot mention that the utility function is a product* of contributions, - you cannot mention that the shape of the utility function is a beta cdf, - you cannot use the words "alpha" or "beta" in your answer. You can, however, - reveal the relative importance of metrics based on the values of alpha and beta, - state qualitatively that even one outcome value close to zero has a big negative effect on your overall satisfaction.
> >
>
> While we cannot guarantee that prompt-based guardrailing is entirely leak-free, our post hoc qualitative analysis of the simulator’s responses indicates that these measures are effective in preventing explicit leakage of the utility function.
>
> Finally, we would like to emphasize that the core contribution of our work is a framework capable of handling unconstrained natural language feedback within the BO loop. LILO is designed to be flexible with respect to the specificity or vagueness of feedback, which we see as an additional advantage of working with NL feedback. In particular, if in a real-world setting the DM actually does have an explicit utility function or at least components thereof available (e.g. in the form of constraints or thresholds), providing those to LILO is “fair game” can substantially improve performance over an interface that relies on, say, pair-wise feedback.

---

> ### Author Response · Authors · 2025-11-22
>
> **W3. Short horizons and Labelling costs.**
>
> We have included additional experiments with larger numbers of trials in the newly added Appendix C.7. and LILO consistently performs competitively compared to the baselines. Note that we are concerned primarily with settings where configurations are very costly to evaluate, and therefore in practice the number of trials is very limited (typically a handful in most of our settings). We would also like to highlight that the goal of preference/prior information elicitation in our setup is to reduce human effort during optimization, and so we study the impact of the queries after only a limited number of batched rounds. In our setup, the experiments are performed in batch (i.e., each trial contains multiple points to be evaluated), hence the sheer number of evaluated candidates is comparable to fully sequential experiments running for many iterations.
>
> Regarding the “unusually large annotation budget relative to very few optimization steps”, we would like to note that the goal of this work, alongside minimization of the experimental evaluations, is the minimization of the interaction cost with the human decision maker – we consider the cost of LLM labelling to be negligible in comparison to the cost of querying the human for feedback. In the new Appendix C.6., we demonstrate that LILO after obtaining just a few messages from the DM is able to perform pairwise labelling with an accuracy above 80%. This means that the time-consuming job of labelling outcomes by human decision makers can be effectively automated with an adequately prompted LLM. In Appendix C.3., we also demonstrate that LILO with just a single question per iteration can match or exceed performance of true utility BO and preferential BO baselines with as many as 8 or 16 queries per iteration, especially in the initial optimization rounds. This suggests that in fact LILO can be more advantageous in feedback-scarce settings.
>
> **W4. Missing baselines.**
>
> We appreciate these additional baselines the reviewer has pointed out. However, the mentioned papers are not necessarily applicable/fair baselines or are already covered by our newly added experiments. Embedding-based surrogates such as mentioned in [1] require non-trivial number of training points, which we do not assume to be available in our setup; Top-k ranking as mentioned in [2] is equivalent to multiple pairwise comparisons, and as our additional experiments in Appendix C.3. show, natural language feedback in LILO can be more sample-efficient than 8 or 16 pairwise comparisons, especially early on in the optimization where the total number of feedback queries is limited. Finally, in our work, instead of exploring the Pareto front of multiple objectives, we assume there is a latent utility function to be learned. This  utility function in fact always exists under moderate regularity conditions according to Debreu's representation theorem. This setting makes it incompatible to compare against multi-objective optimization methods such as [3].
>
> **W5. Novelty.**
>
> Thank you for the opportunity to clarify our core contributions. As discussed in our literature review, many prior works leverage LLMs to directly generate candidates for optimization, despite LLMs’ limitations in probabilistic reasoning and uncertainty modeling. In contrast, our approach deliberately restricts the LLM’s role to a simple function of translating the decision maker’s goals and domain knowledge into pairwise preference labels, which serve as an optimizable numerical signal for Bayesian optimization. Crucially, our framework retains the principled foundations of BO—such as probabilistic GP surrogates and acquisition functions—rather than delegating these components to the LLM. We believe this integration is both organic and efficient, as evidenced by our experimental results. In the context of related work, we see the careful placement of the LLM within the BO loop as a meaningful contribution, offering a practical and robust way to incorporate human goals and knowledge into BO.
>
> ### Questions
>
> See our response to W2.
>
> ---
>
> Thank you for your valuable feedback, which helped us improve the paper.  We hope our answer resolves your concerns and we are eager to engage in further discussion to address any remaining questions.

---

### Official Review · Reviewer_yRqr · 2025-11-11

**Soundness:** 2
**Presentation:** 2
**Contribution:** 2
**Rating:** 4
**Confidence:** 4

**Summary:**

This paper addresses the human-in-the-loop Bayesian optimisation problem by introducing large language models (LLMs) as a user-friendly interface to facilitate and accelerate preference learning. Two surrogate models are proposed that seperately capture the mappings from inputs and  outputs to the preference space. A batched acquisition function is employed to generate evaluation candidates, after which the LLM elicits user preferences and potential narrative explanations. Another acquisition function selects the top-K informative pairwise comparisons, with preference labels inferred from the LLM ouputs. Across multiple benchmarks tasks, the proposed approach demonstrates competitive performance compared with classical BO, preference-based BO, and LLM-assisted optimisation baselines.

**Strengths:**

**Significance:** The topic is increasingly important. Human-in-the-loop optimisation represents a crucial direction for advancing domains such as scientific discovery (aligning with expert knowledge) and healthcare (addressing individual needs).

**Clarity:** The paper is clearly written and structured. The proposed approach is presented with sufficient detail and generality, making it adaptable to other optimisation frameworks. The illustrations (e.g., Figure 2, Example 1, Figure 3) effectively support understanding of the methodology.

**Orignality:** The introduction of a two-surrogate framework that leverages LLMs both as an interactive interface and as preference labellers appears original.

**Weaknesses:**

The contribution of the paper appears limited:
1. Most algorithmic components, including the Gaussian process (GP) models and acquisition functions, are well established in prior work, as acknowledged in the paper.
2. The proposed integration framework lacks sufficient technical discussion or analysis regarding its robustness and convergence properties. In particular, the method may be fragile to biased or inconsistent LLM outputs, and it is unclear whether the optimisation reliably converges to high-utility solutions.
3. As the study involves multi-dimensional outputs, it remains uncertain whether LLMs can effectively handle multi-objective scenarios, such as those requiring Pareto-optimal trade-offs or modeling of decision-makers' stochastic/utopian preferences [1].

The experiment assessment is underdeveloped: The choice of limiting each run to only eight iterations is not well justified. Either a large number of iterations or an explicit analysis of the convergence gap to the optimal utility value would be necessary to demonstrate whether the method achieves a reasonable exploration-exploitation balance (an essential principle of Bayesian optimisation) rather than becoming trapped in suboptimal regions.

The experimental comparisons may not be entirely fair: For instance, in one query (line 227, Example 1), the decisionmaker (DM) is required to prioritise multiple metrics simultaneously. This imposes a substantially higher cogvitive load than a standard pairwise comparison. The paper should therefore account for DM workload beyond simple query counts to ensure a balanced and meaningful comparison between methods.

[1] Direct Preference-Based Evolutionary Multi-Objective Optimization with Dueling Bandits, NeurIPS 2024.

**Questions:**

1. What is the key distinguishing aspect from prior work?

2. How can a fairer experimental comparison be achieved in terms of DM workload?

3. Could you include experiments with a larger number of iterations and report the optimal utility values?

4. How does the choice of $B^{\text{pf}}$ affect performance on other tasks beyond the numerical DTLZ2 benchmark (Figure 6)?

---

> ### Author Response · Authors · 2025-11-22
>
> Thank you for your feedback. We appreciate the time and effort you put into assessing our paper. Below, we address your questions and comments point by point.
>
> ---
>
> ### Weaknesses
>
> **W1. Contributions of our work.**
>
> As we discuss in our literature review, many prior works leverage LLMs to directly generate candidates for optimization, despite LLMs’ limitations in probabilistic reasoning and uncertainty modeling. In contrast, our approach deliberately restricts the LLM’s role to a simple function of translating the decision maker’s goals and domain knowledge into pairwise preference labels, which serve as an optimizable numerical signal for Bayesian optimization. Crucially, our framework retains the principled foundations of BO—such as probabilistic GP surrogates and acquisition functions—rather than delegating these components to the LLM. We believe this integration is both organic and efficient, as evidenced by our experimental results. In the context of related work, we see the careful placement of the LLM within the BO loop as a meaningful contribution, offering a practical and robust way to incorporate human goals and knowledge into BO.
>
> **W2. Robustness and Convergence.**
>
> Thank you for raising this important point. We agree that introducing an LLM into the BO loop adds an element of unpredictability. However, this comes with meaningful benefits: natural-language feedback is often (a) substantially more information-rich and (b) far more natural and accessible for human decision makers. Any method that leverages LLMs for optimization necessarily inherits this “black-box” aspect, and this concern applies broadly across recent work in the area.
>
> Crucially, unlike prior approaches that rely on LLMs to perform all core components of BO (e.g., candidate generation, utility estimation, acquisition decisions), our framework strictly limits the role of the LLM to translating the DM’s natural-language feedback into a quantitative optimization signal. All subsequent components—utility modeling via a GP surrogate and candidate selection via a principled acquisition function—remain fully probabilistic and interpretable.
>
> As demonstrated empirically, LILO consistently identifies high-utility solutions within only a few experimental trials and exhibits substantially better stability and convergence than the two LLM-centric baselines considered. We view this careful placement of the LLM within the BO pipeline as one of our key contributions: it offers a practical and robust mechanism for incorporating human goals and domain knowledge into BO while maintaining the methodological rigor of conventional surrogate-based optimization.
>
> Finally, to provide additional insights into the internal mechanics of LILO we have provided a new set of analyses. See paragraphs *Additional insights* in the main rebuttal and Appendix C.6. of the updated submission. We demonstrate that LILO is able to faithfully recover the ground-truth utility function’s shape, even just after a few experimental trials. Furthermore, we show that the LLM’s accuracy in generating pairwise labels for utility estimation is very high and steadily increases with each round of experimentation and feedback collection.
>
> **W3. Multi-objective optimization**
>
> Firstly, we would like to clarify that all utility functions considered in this work are indeed multi-objective – inducing varying preference profiles and trade-offs between all dimensions of the outcome space (we consider test problems with up to 9 outcome variables). Regarding the setup of stochastic/utopian preferences, it would present a challenge universally impacting all preference-learning-based algorithms, without further assumptions being made. With regards to Pareto-optimal trade-offs, Debreu's representation theorem shows that under moderate regularity assumptions, any well-defined preference ordering over Pareto-optimal outcomes admits a utility representation, which is intended to be learned via LLM’s in-context learning in LILO.
>
> **W4. Limited number of trials.**
>
> See the paragraph *Longer horizons* in the main rebuttal for a detailed response including new results of LILO with more experimental rounds (Appendix C.7. of the updated submission).

---

> ### Author Response · Authors · 2025-11-22
>
> ### Questions
>
> **Q1.**  See our response to W1.
>
> **Q2.** See the paragraph *Fairness of comparison* of the main rebuttal for a detailed discussion of this topic including new experimental results of LILO compared to baselines with larger batch sizes.
>
> **Q3.** Yes, however this is not the focus of this paper. Please consult the paragraph *Longer horizons* in the main rebuttal for a detailed response including new results of LILO with more experimental rounds (Appendix C.7. of the updated submission).
>
> **Q4.**  Newly added results in Appendix C.3. demonstrate a companion's of LILO with $B^{\text{pf}} \in \\{1, 2\\}$ against the quantitative baselines with $B^{\text{pf}} \in \\{1, 2, 4, 8, 16\\}$.
>
> ---
>
> Thank you for your valuable feedback, which helped us improve the paper.  We hope our answer resolves your concerns and the additional experiments presented further strengthen our contribution justifying a potential revision of the paper's score.

---

### Official Review · Reviewer_ZCQ2 · 2025-11-11

**Soundness:** 3
**Presentation:** 3
**Contribution:** 3
**Rating:** 6
**Confidence:** 3

**Summary:**

This paper addresses the problem of optimizing the outputs of a black-box system with respect to a decision maker's (DM) preferences. The authors propose LILO, a hybrid framework that leverages large language models (LLM) to extract utility information from free-form textual feedback and principled Bayesian Optimization (BO) for candidate generation. Specifically, the utility estimation is achieved via training GP-based proxy utility models on LLM-generated pairwise comparisons. The method is empirically validated on synthetic and real-world test functions subject to various utility functions, demonstrating on-par or improved performance over LLM-based and GP-based baselines.

Overall, this work is well-motivated and the paper is generally well-written. Given clarification of some empirical concerns, I am willing to increase the score.

**Strengths:**

1. **Strong motivation:** The proposed method provides a DM-friendly and powerful solution to black-box optimization with subjective DM perferences. By introducing LLMs, it enables flexible, free-form DM feedback and the incorporation of different types of domain priors.
2. **Empirical performance:** The results show that LILO achieves on-par or superior performance against both LLM-based and GP-based baselines on synthetic and real-world test functions across various utility functions. The current design is well-supported by a set of ablation study.

**Weaknesses:**

The process of the LLM learning the utility given DM's natural language feedback and experimental data introduces a black-box mapping with unclear internal mechanism and implicit priors. This impacts the principled nature of the overall optimization.

**Questions:**

1. Larger number of trials ($T$): The current results are limited to short horizion, $T=8$. Experimenting with a larger number of trials for LILO and the baselines would provide a more complete understanding of the method's convergence properties and sample efficiency.
2. Figure 3: Could you explain the performance discrepancy in the LLM baselines for Thermal Comfort Type A and Type B?
3. Line 475 - 476, "In our experiments, we found this effect to be manageable by ablating over a variety of utility functions and test problem combinations": Could you elaborate a bit more on the issue this statement refers to, and how you address it through ablating over utility functions and test problems?
4. Would be valuable to visualize the estimated utility values.
5. Minor comments: Figure 2 notation. In the Candidate Generation & Experimentation panel, the batch size of canidates should be denoted as $B^{\text{exp}}$ instead of $B^{\text{pf}}$?

---

> ### Author Response · Authors · 2025-11-22
>
> We thank the reviewer for their constructive and insightful feedback. Your comments helped us clarify key contributions of our work and motivated several improvements to the paper. Below we address the reviews concerns and questions point by point.
>
> ---
>
> ### Weaknesses
>
> **Black-box LLM and the principled nature of optimization.** Thank you for raising this important point. We agree that introducing an LLM into the BO loop adds an element of unpredictability. However, this comes with meaningful benefits: natural-language feedback is often (a) substantially more information-rich and (b) far more natural and accessible for human decision makers. Any method that leverages LLMs for optimization necessarily inherits this “black-box” aspect, and this concern applies broadly across recent work in the area.
>
> Crucially, unlike prior approaches that rely on LLMs to perform all core components of BO (e.g., candidate generation, utility estimation, acquisition decisions), our framework strictly limits the role of the LLM to translating the DM’s natural-language feedback into a quantitative optimization signal. All subsequent components—utility modeling via a GP surrogate and candidate selection via a principled acquisition function—remain fully probabilistic and interpretable.
>
> As demonstrated empirically, LILO consistently identifies high-utility solutions within only a few experimental trials and exhibits substantially better stability and convergence than the two LLM-centric baselines considered. We view this careful placement of the LLM within the BO pipeline as one of our key contributions: it offers a practical and robust mechanism for incorporating human goals and domain knowledge into BO while maintaining the methodological rigor of conventional surrogate-based optimization.
>
> Finally, to provide additional insights into the nternal mechanics of LILO we have provided a new set of analyses. See paragraphs *Additional insights* in the main rebuttal and Appendix C.6. of the updated submission. We demonstrate that LILO is able to faithfully recover the ground-truth utility function’s shape, even just after a few experimental trials. Furthermore, we show that the LLM’s accuracy in generating pairwise labels for utility estimation is very high and steadily increases with each round of experimentation and feedback collection.
>
> ### Questions
>
> **Q1. Larger number of trials**
>
> See paragraph *Loner horizons* of the main rebuttal and the newly added Appendix C.7.
>
> **Q2. Performance discrepancy for Thermal Comfort Type A and B.**
>
> As described in Appendix B.2, Thermal Comfort Type A and B are designed to simulate the distinct preference of two individuals. Type A is designed to simulate preferences of an office worker in light clothing and a moderate tolerance for undesirable conditions while Type B is designed to simulate preferences of a summer athlete wearing light sport kit, with a lower tolerance for adverse conditions. Because of the more strict and specific requirements for a comfortable environment in Type B, all methods (except for the LLM baselines) start from a lower starting utility and end at a lower final utility at the end of the optimization. LLM baselines performed well in the initial guess for both Type A and Type B environments, generating points with a utility of around 0.8. Likely, the specificity and strictness of these utility functions, especially Type B utility, provided sufficient context for such LLM guessworks to work well.
>
> **Q3.**  *"In our experiments, we found this effect to be manageable by ablating over a variety of utility functions and test problem combinations"*
>
> The issue described here refers to the potential concern that the LLM model used by LILO to perform pairwise comparisons may not be able to accurately learn the underlying utility function and, as a result, hurt the optimization quality.  We have experimented with a variety of utility functions and demonstrated that LILO is able to perform robustly across all of them.
>
> **Q4. Visualizations**
>
> Thank you for this suggestion. In the newly added Appendix C.6., we demonstrate that the fitted utility functions reflect the relevant properties of the ground-truth utilities. In addition we also show that the LLM’s accuracy in performing pairwise preference labelling consistently increases as we gather more feedback in natural language.
>
> **Q5. Typo**
>
> We thank the reviewer for pointing out the typo. We have updated this in the revised manuscript accordingly.
>
> ---
>
> We hope these clarifications and additions address the reviewer’s concerns. We believe that the additional experiments enhance the significance of our findings. We are grateful for the reviewer’s suggestions.

---

### Author Response · Authors · 2025-11-22
**Main Rebuttal: Common Concerns and Summary of Revisions**

Dear Reviewers,

We would like to express our gratitude to all the reviewers for their constructive feedback and insights on our submission*.* We appreciate the opportunity to clarify and enhance our manuscript based on the feedback received.

Below we outline the key actions taken in response to common concerns among the reviewers. Remaining queries of individual reviewers have been addressed in personalised responses.

---

## New Results

### Fairness of comparison

Reviewers raised concerns about the fairness of comparing LILO—where the DM provides ($B^{\text{pf}}$) natural-language responses per trial—with the quantitative baselines (true utility BO and preferential BO), which each receive the same number of ground-truth utility queries or pairwise comparisons, respectively. It is, however, not straightforward to compare the DM's workload necessary to answer natural language questions vs. the mental effort required to evaluate outcomes on an absolute scale or provide pairwise comparisons. The DM’s burden depends on the specific questions being asked, and for conventional feedback formats it depends on the complexity of the underlying latent utility function and the types of outcomes being judged.

To better address this concern, we have added a new set of experiments (Appendix C.3) that compare LILO with $B^{\text{pf}} \in \\{1,2\\}$ against the quantitative baselines using $B^{\text{pf}} \in \\{1,2,4,8,16\\}$. These experiments aim to empirically evaluate how many pairwise comparisons or ground-truth utility queries are roughly equivalent to a single natural-language message from the DM. Across several environments, we observe that even a single natural-language statement can match—or in some cases outperform—as many as 8 or 16 pairwise comparisons or pointwise utility evaluations. The advantage of LILO is most pronounced in the early stages of experimentation, highlighting the high information density and sample efficiency of natural-language feedback, when used effectively.

Finally, we emphasize that both quantitative baselines rely on types of feedback that are unrealistic in real-world human-in-the-loop optimization. The ground-truth utility function is never directly observable, as it reflects an internal latent preference model. Likewise, the preferential BO baseline assumes noise-free comparisons, whereas human comparisons invariably include cognitive noise and stochasticity—factors that would further degrade its real-world performance. Thus, while our expanded experiments address the fairness question empirically, they also reinforce that the quantitative baselines set a strong benchmark compared to what is feasible with actual human decision makers.

### Additional insights

In response to concerns about the black-box nature of using LLMs for utility estimation within BO, we conducted two additional analyses: (i) an evaluation of the LLM’s accuracy in pairwise preference labeling, and (ii) a qualitative assessment of the learned utility function’s shape. These results are now included in the newly added Appendix C.6. In summary:

- **High accuracy in LLM-based preference labeling.**

    During step 3 of LILO, the LLM is tasked with generating pairwise preference labels. We find that its accuracy is remarkably high: already at the very first trial, the LLM’s pairwise choices agree with those implied by the ground-truth utility function over 80% of the time. This accuracy steadily improves over subsequent trials, reaching approximately 90%. These findings indicate that the labor-intensive process of labeling outcomes—which would otherwise require substantial effort from human decision makers—can be reliably offloaded to a properly prompted LLM. LILO relies on this property to translate natural-language feedback into a high-fidelity optimization signal suitable for principled BO with GPs.

- **Faithful reconstruction of the utility landscape.**

    We further demonstrate that the utility function fitted via LLM-generated labels and subsequent GP modeling captures key structural properties of the underlying ground-truth utility. For instance, in the L1 utility setting, even in the earliest trials, the peaks of the learned utility functions align closely with those of the true utility. This provides qualitative evidence that the LLM+GP yields a utility approximation whose shape meaningfully reflects the true latent preferences.

We hope that these additional analyses provide deeper visibility into the internal mechanics of LILO and help to alleviate concerns about the methodological soundness of our approach.

---

> ### Author Response · Authors · 2025-11-22
> **Main Rebuttal: Common Concerns and Summary of Revisions (continued)**
>
> ### Longer horizons
>
> Some reviewers asked about LILO’s performance over longer optimization horizons. We would like to note that this work focuses on settings where evaluating each configuration is costly, and therefore, the total number of trials is inherently limited. In addition to reducing experimental costs, a key objective of our approach is to minimize human effort during optimization, which motivates our emphasis on the impact of only a few batched feedback rounds. Importantly, because each trial evaluates multiple candidates in parallel, the total number of evaluated points is comparable to many-iteration fully sequential experiments.
>
> Nevertheless, to address the reviewers’ question directly, we have added new experiments evaluating LILO over longer horizons—up to 16 batched trials (Appendix C.7). We find that LILO maintains competitive performance even in this extended regime. Consistent with our main results, the relative advantage of LILO diminishes as the number of trials increases: quantitative baselines eventually catch up and, in some cases, surpass LILO in the long run. This behavior is expected—LLMs’ in-context learning capabilities, while powerful, are inherently limited and do not provide the convergence guarantees available to conventional BO methods that operate on quantitative feedback formats.
>
> ---
>
> We are grateful for the reviewers’ feedback, which helped us clarify the key contributions of our work and strengthen our evaluation. We welcome further discussion to address any remaining questions and look forward to the opportunity for our study to make a meaningful contribution to the community.
>
> Kind regards,
>
> The authors

---

### Meta-Review · Area_Chair_drqa · 2026-01-07

**Summary:**

This is a boarderline paper. The main concerns from the reviewers are 1) the limited novelty (since both GP BO and LLM as preference judge are well studied, though not together)  2) insufficient experimental results (due to limited optimization iterations, dated benchmarks, missing high-dim or modern HPO benchmarks, lack of other LLM BO or optimization baselines). There are two reviewers that are outliers: one review is exceptionally short and vague, and the other does not touch the core part of the paper. However, even if we down weight them, the rest reviewers are not unanimous. Overall, while I think this paper has its merits, given the nature of the paper having limited novelty on the algorithmic side and lacking theoretical analyses, I would expect more results on the empirical side to supplement. Nonetheless, I do not fully disregard the reviewers' points on limited experiments, especially on the dated benchmarks and lack of baselines. Therefore, I would not recommend acceptance of the current version.

**Reviewer Concerns:**

See summary.

**Reviewer Scores:**

Reviewer ZCQ2 seemed to already changed the score.
For the rest, I suspect the scores will remain similar, as none seems very excited about the paper.

---

### Decision · Program_Chairs · 2026-01-26

Reject